# Adaptive Quasimetric Mapping : Principled Topological Abstraction for Robust Offline Goal-Conditioned Navigation

**Anthony Kobanda** [1 2]   **Waris Radji** [1]   **Odalric-Ambrym Maillard** [1]   **Rémy Portelas** [2]

## Abstract

Goal-Conditioned Reinforcement Learning aims to design agents that can reach specified goals, notably from previously collected trajectories in the offline setting. In this context, graph-based approaches have been proposed to mitigate value-estimation errors in long-horizon navigation tasks. However, existing approaches typically rely on dense keypoint coverage of the dataset support, resulting in computationally expensive planning. Moreover, they lack an explicit mechanism to adapt to topological changes (e.g., new obstacles), hindering deployment in live applications such as video games. To address these shortcomings, we propose **Adaptive Quasimetric Mapping** (**AQM**), an offline framework leveraging a "time-to-reach" quasimetric learned from the available data. Crucially, it builds a sparse cover of the dataset support, as a greedy approximation to a dominating set problem. At test-time, the resulting graph is carefully pruned by comparing the observed edge traversal time against a time-to-reach budget derived from the quasimetric, thus enabling zero-shot replanning. Empirically, we evaluate AQM on navigation tasks ranging from a classical to a video-game-like benchmark evaluating adaptation across tasks. We show that AQM achieves competitive performance while requiring up to $100\times$ fewer keypoints than prior approaches, demonstrating the relevance of principled topological abstraction for robust offline goal-conditioned navigation.

## 1. Introduction

Humans improve knowledge and master complex skills by iteratively exploring and refining different internal notions of proximity ("*How far am I from succeeding ?"*) and progress ("*Is this action helpful ?*") (Ericsson et al., 1993). Reinforcement learning (RL) (Sutton & Barto, 2018) offers the computational analogue, combining trial-and-error with value function approximation to learn algorithms and agents to solve sequential decision problems, and is extended to high-dimensional perception planning, and control with Deep Reinforcement Learning (Arulkumaran et al., 2017).

In many real-world scenarios, such as creating intelligent Non-Player Characters (NPCs) in large-scale video games, active exploration is costly (Biré et al., 2025), while vast amounts of logged interactions exist. Offline RL improves data efficiency by learning from pre-collected trajectories (Levine et al., 2020). Nevertheless, without the ability to query states and evaluate new actions, learning agents must optimize their policy beyond the dataset's behavior distribution (Peng et al., 2019; Kostrikov et al., 2022), a process that is sensitive to bootstrapping errors, particularly in long-horizon settings (Park et al., 2024a). Moreover, once the data has been recorded, overcoming changes in the environment, whether due to evolving dynamics or design updates, poses additional challenges for the learned policy.

By conditioning an agent's policy on a specific target, Goal-Conditioned RL (GCRL) repurposes decision problems as reachability tasks (Kaelbling, 1993; Liu et al., 2022). Although techniques such as Hindsight Experience Replay (Andrychowicz et al., 2017) mitigate sparse rewards and Hierarchical Policies (Li et al., 2022; Park et al., 2023) ease path planning, most fall short when tackling long-horizon navigation tasks, which is a focus of our research. Recent GCRL benchmarks such as **OGBench** (Park et al., 2025a) highlight these challenges in diverse experimental settings.

To tackle these issues, a promising route is to plan through learned latent representations that encode reachability. Contrastive Learning algorithms (Eysenbach et al., 2022) cluster similar states but learn uncalibrated similarities rather than true distance metrics, a key challenge highlighted by recent work on temporal distance functions. Quasimetric RL provides such a reachability-aware distance, however resulting policies fail on complex and long-horizon tasks (Wang et al., 2023). State-space mapping algorithms, such as Hilbert Representations (Park et al., 2024b) and then Graph-Assisted Stitching (Baek et al., 2025), improve long-horizon navigation, but their strongest operating points can require large graphs, making inference costly. Moreover, these methods do not adapt to changes during evaluation.

To address these issues, we introduce a new framework : **Adaptive Quasimetric Mapping (AQM)**, thoroughly laid out in *Figure 1*. It is adaptive in the following sense: by monitoring a time-to-reach budget between keypoints during execution, an agent can infer geometric inconsistencies, trigger unreliable edges invalidation, update the initial graph, and plan new paths without having to retrain a new policy.

[1]Inria, Univ. Lille, CNRS, Centrale Lille, UMR 9198-CRIStAL, France [2]Ubisoft La Forge, Bordeaux, France. Correspondence to : Anthony Kobanda .

*Proceedings of the $43^{rd}$ International Conference on Machine Learning*, Seoul, South Korea. PMLR 306, 2026. Copyright 2026 by the author(s).

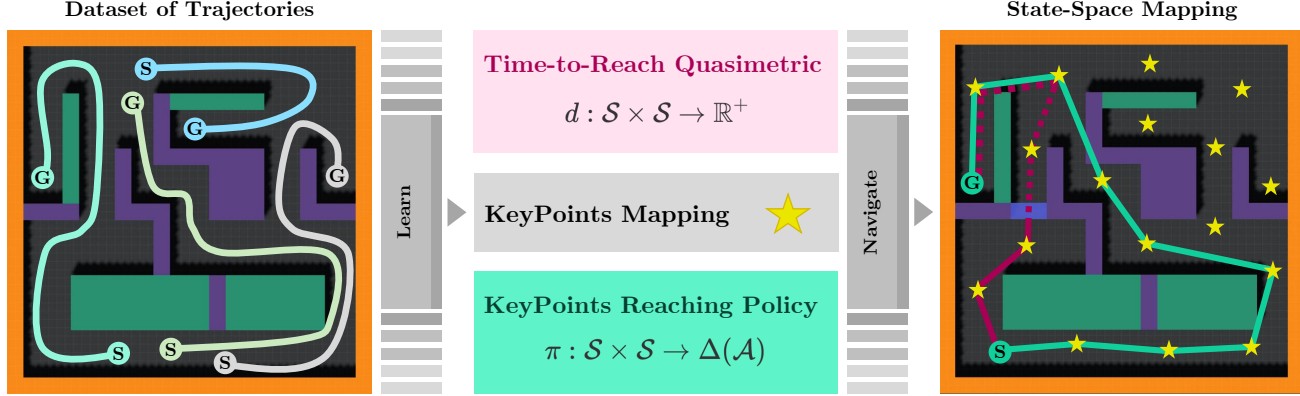

*Figure 1.* **Adaptive Quasimetric Learning & Mapping (AQM)**. **(left)** From a dataset of trajectories $\mathcal{D} = \{\tau_i = (s_0^{(i)}, a_0^{(i)}, \ldots, s_T^{(i)})\}$ ; **(middle)** We learn a time-to-reach quasimetric $d : \mathcal{S} \times \mathcal{S} \rightarrow \mathbb{R}^+$, a goal-conditioned policy $\pi : \mathcal{S} \times \mathcal{S} \rightarrow \Delta(\mathcal{A})$ ; **(right)** The learned quasimetric is used to build a sparse topological graph of the state space, with keypoints (⭐) providing near uniform coverage of the dataset support. During inference, AQM plans a path from a starting position (S) to a goal (G) and follows it using the learned policy. It can adapt to topological changes at test-time, such as a **blocked path**, by monitoring budget overruns given predicted time-to-reach values. Consequently, AQM performs zero-shot replanning, eventually backtracking, to reach the goal through another **feasible path**.

We argue that a bottleneck in current graph-based navigation lies in the reliance on uncalibrated distance heuristics, which dense keypoint sets to ensure connectivity. In contrast, we establish a principled approach: by learning a geometrically consistent time-to-reach quasimetric, we are able to extract parsimonious graphs as topological abstractions that are both computationally efficient and inherently adaptive.

Our main contributions are as follows :

(1) **Geometric Foundation :** We leverage a time-to-reach quasimetric function as a principled tool for topological abstraction, serving as geometric prior for navigation, and enabling to build a connectivity graph and to plan ;

(2) **Topological Abstraction :** We propose a Keypoint Selection Strategy inspired by dominating set theory, providing covering and connectivity analysis under standard density assumptions, yielding sparse graphs ;

(3) **Test-Time Adaptation :** We introduce an Adaptive Planning Mechanism. By monitoring the time-to-reach quasimetric, we flag inconsistencies at test-time, set unreliable edges as invalid, and replan alternative paths ;

(4) **Empirical Validation :** We evaluate our framework on two offline navigation benchmarks, demonstrating competitive performance, while improving scalability relative to recent graph-based approaches.

While *Section 2* reviews the relevant and related literature, *Section 3* presents the necessary theoretical background contextualizing our research. In *Section 4*, we detail our approach, first going through an algorithmic overview, then focusing on the learning process and the inference pipeline. *Section 5* presents our experimental setting, the evaluation protocol, the main results obtained and ablation studies. We conclude in *Section 6*, notably with the limitations of our work, experiments, and currently identified future work.

## 2. Related Work

**Offline Goal-Conditioned Reinforcement Learning.** In the Goal Conditioned RL framework (Kaelbling, 1993), states are augmented with goals, and policies are optimized to reach them. In the offline setting (Levine et al., 2020), learning must remain within the support of a given dataset, while aiming at surpassing the behavioral policy. Although value-based methods help learning (Kostrikov et al., 2022), they can fail on long-horizon tasks due to approximation errors in learned value functions (Park et al., 2024a; 2025b).

**Hierarchical Planning & Policy Distillation.** Hierarchical approaches mitigate the aforementioned issue by planning (Li et al., 2022; Park et al., 2023) from a high-level policy, and learning actions to closer goals through a low-level one. Policy distillation methods like SAW (Zhou & Kao, 2025) compress this hierarchical structure into a flat one, offering improved stability and computational efficiency compared to two-level policies. Nevertheless, even for such methods, stitching long trajectories and handling changes at test-time remains difficult (Park et al., 2025a; Kobanda et al., 2025).

**Representation Learning.** A key idea in order to improve performance in RL is to learn a structured latent projection or representation of the state space (Fuest et al., 2024; Echchahed & Castro, 2025). Many approaches focus on learning behavioral similarity through self-supervised and contrastive learning objectives. Among these, CURL (Laskin et al., 2020) and CRL (Eysenbach et al., 2022) aim to map temporally close states to nearby points in a learned latent space. Other methods learn explicit latent dynamics-aware models (Gelada et al., 2019; Shi et al., 2023), aiming to capture the transition structure of the given environment. While these approaches have valuable use cases, they do not produce calibrated measure of time-to-reach between states, which we argue is a key component for principled GCRL.

**Distance Learning.** Although value functions implicitly encode reachability, they are poor tools to qualitatively measure reachability between states. This has motivated distance learning, with a key insight that such a function induces an asymmetric cost-to-go or a *quasimetric*. Interval Quasimetric Embeddings (IQE) (Wang & Isola, 2022) is an expressive basis to learn such functions. Its direct successor, Quasimetric RL (QRL) (Wang et al., 2023) established a relevant principled learning paradigm in RL. An alternative is to enforce self-consistency through Bellman updates. For instance, Deep Floyd-Warshall RL (Dhiman et al., 2018) translates the triangle inequality into an auxiliary loss for a goal-conditioned critic, similarly to Transitive RL (TRL) (Park et al., 2025c) which ease bootstrapping. An other approach consists in leveraging temporal-difference learning like in the Minimum Action Distance framework (MAD) (Steccanella et al., 2025). Alternatively, Temporal Metric Distillation (TMD) (Myers et al., 2025) blends a contrastive objective with auxiliary losses enforcing metric consistency.

**Topological Abstraction.** A strategy for GCRL is to build sparse mappings (Huang et al., 2019). Early approaches used pre-computed distances to fixed points and navigated via hops, but suffer coverage gaps or heavy nearest-neighbor search (Savinov et al., 2018; Kim et al., 2021). Recent work scale such methods with learned temporal distances. HILP (Park et al., 2024b) builds a Hilbert-space representations but may be memory-intensive. such as QPHIL (Canesse et al., 2025) ease planning by dividing the state space, but underperform due to out-of-distribution trajectory predictions. Among these, Graph-Assisted Stitching (GAS) (Baek et al., 2025) represents the state-of-the-art approach. It builds a graph from data, achieving strong performance, while avoiding out-of-distribution subgoals. Nevertheless, HILP and GAS rely on a large number of nodes at inference.

**Adaptive Navigation.** Recent frameworks also study how adaptive planning structures can be leveraged. For example, Breadth-First Exploration on Adaptive Grid for RL (BEAG) (Yoon et al., 2024), uses subgoals on an adaptively refined grid to guide exploration. This direction is complementary to AQM, as BEAG targets online exploration, whereas AQM targets goal-conditioned navigation from pre-collected data.

## 3. Preliminaries

We consider a **Markov Decision Process (MDP)** as tuple $\mathcal{M} = \big( \mathcal{S}, \mathcal{A}, \mathcal{P}_{\mathcal{S}}, \mathcal{P}_{\mathcal{S}}{}^{(0)}, \mathcal{R}, \gamma \big)$, which provides a formal setting for RL, where $\mathcal{S}$ is a state space, $\mathcal{A}$ an action space, $\mathcal{P}_{\mathcal{S}} : \mathcal{S} \times \mathcal{A} \to \Delta(\mathcal{S})$ a transition function, $\mathcal{P}_{\mathcal{S}}{}^{(0)} \in \Delta(S)$ an initial distribution over the states, $\mathcal{R} : \mathcal{S} \times \mathcal{A} \times \mathcal{S} \to \Delta(\mathbb{R})$ a reward function, generally deterministic, and $\gamma \in\ ]0, 1]$ a discount factor. An agent's behavior follows a policy $\pi_\theta : \mathcal{S} \to \Delta(\mathcal{A})$, parameterized by parameters $\theta \in \Theta$. In Reinforcement Learning, we aim for optimal parameters $\theta_{\mathcal{M}}^*$ maximizing the expected cumulative reward $J_{\mathcal{M}}(\theta)$.

**Offline Goal-Conditioned RL (GCRL)** extends the MDP to include a goal space $\mathcal{G}$, introducing $\mathcal{P}_{\mathcal{S},\mathcal{G}}{}^{(0)}$ an initial state and goal distribution, $\phi : \mathcal{S} \to \mathcal{G}$ a function mapping each state to the goal it represents, and $d : \mathcal{G} \times \mathcal{G} \to \mathbb{R}^+$ a distance metric on $\mathcal{G}$. The policy $\pi_\theta : \mathcal{S} \times \mathcal{G} \to \Delta(\mathcal{A})$ and the reward function $\mathcal{R} : \mathcal{S} \times \mathcal{A} \times \mathcal{S} \times \mathcal{G} \to \mathbb{R}$ are now conditioned on a goal $g \in \mathcal{G}$. In our experimental setup, we consider sparse rewards allocated when the agent reaches the goal within a range $0 \leq \epsilon : \mathcal{R}(s_t, a_t, s_{t+1}, g) = \mathbb{1}\big( d(\phi(s_{t+1}), g) \leq \epsilon \big)$. Given a dataset of trajectories $\mathcal{D} = \big\{ (s, a, r, s', g) \big\}$, the policy loss is optimized to reach the specified goals. This formulation represents the core problem we are addressing.

Unlike metrics, which are symmetric, GCRL value (or time-to-reach) functions are generally asymmetric: reaching A from B can require a different expected cost than reaching B from A. This property aligns with **Quasimetric** functions (Wilson, 1931) $d : \mathcal{G} \times \mathcal{G} \to \mathbb{R}^+$ satisfying : *Non-negativity* : $d(x, y) \geq 0$ ; *Identity of indiscernibles* : if $d(x, y) = 0$ then $x = y$ ; *Reflexivity* : $d(x, x) = 0$ ; *Triangle inequality* : $d(x, z) \leq d(x, y) + d(y, z)$ ; but not necessarily symmetry of the distance : $d(x, y) \neq d(y, x)$ (whereas **Metrics** do).

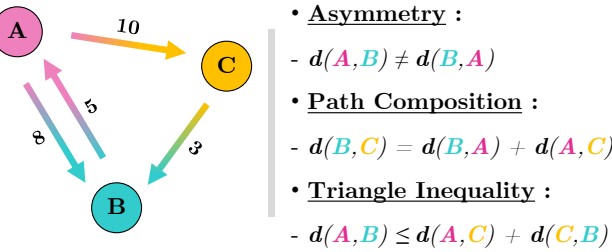

- **Asymmetry :**
  - $d(A, B) \neq d(B, A)$
- **Path Composition :**
  - $d(B, C) = d(B, A) + d(A, C)$
- **Triangle Inequality :**
  - $d(A, B) \leq d(A, C) + d(C, B)$

*Figure 2.* **A Brief Anatomy of a Quasimetric Space**.

This self-consistency, defined by the triangle inequality, is the foundation for shortest-path algorithms. In the case of a finite graph, the Floyd-Warshall algorithm provides a classic method for computing an all-pairs shortest-path quasimetric.

**The Floyd-Warshall Algorithm** (Floyd, 1962). Given a directed graph $G = (V, E)$ where $V$ is a finite set of vertices and $w(i, j) \in \mathbb{R}^+ \cup \{+\infty\}$ is the weight of an edge in $E$ from vertex $i$ to $j$, the algorithm finds the shortest path quasimetric $d^*(i, j) \in \mathbb{R}^+ \cup \{+\infty\}$ between all pairs $(i, j)$.

**Proposition 3.1** (Floyd-Warshall Update)**.** *Let's consider $d_k(i, j)$ be the shortest path distance from vertex $i$ to $j$ using only intermediate vertices from the set $\{1, 2, \ldots, k\}$. The algorithm initializes $d_0(i, j) = w(i, j)$ and iteratively computes for $k = 1, \ldots, |V|$ :*

$$d_k(i, j) \leftarrow \min \big( d_{k-1}(i, j),\ d_{k-1}(i, k) + d_{k-1}(k, j) \big)$$

*After $|V|$ iterations, $d_{|V|}$ contains the true all-pairs shortest-path distances, which form a valid quasimetric on the graph.*

This iterative process of finding shorter paths by considering intermediate *subgoals* vertices provides the core inspiration for our replanning and sample-based adaptive mechanism, which is further detailed and explained in *Section 4*.

# 4. Adaptive Quasimetric Learning & Mapping

We propose **AQM**, a framework that aims at unifying geometric representation learning with efficient graph-based planning. We proceed in three parts : (*Section 4.2*) Learning a *time-to-reach* between any states ; (*Section 4.3*) Building a topological graph of keypoints ; (*Section 4.4*) Training a goal-conditioned policy to reach any proximal goal. The complete learning procedure is summarized in *Algorithm 1*.

## 4.1. AQM Learning Algorithm : an Overview

---

**Algorithm 1 AQM Learning Pipeline**

---

**Require:** *Dataset of traces* $\mathcal{D} = \{\tau_i = (s_0^{(i)}, a_0^{(i)}, \ldots, s_T^{(i)})\}$
**Require:** *Batch size* $B$ , *# Epochs* $E$ , *Learning Rate* $\eta$
**Require:** *Minimum Keypoint Separation Distance* $\delta$

**Initialize:** Quasimetric $d_\theta$
**Initialize:** Lagrange Multiplier $\lambda$
**Initialize:** Main & Target Policies $\pi_\psi$ , $\pi_{\psi^-}$

**Notation:** $s_i$ , $s_i'$ a state and an immediate next state
**Notation:** $a_i$ an action performed at $s_i$ leading to $s_i'$
**Notation:** $w_i$ a subgoal at a fixed timestep distance
**Notation:** $g$ a goal sampled from some distribution

1: **for** $t = 1$ to $E$ **do**
2:   **Sample batch** : $\{(s_i, a_i, s_i', w_i, g)\}_{i=1}^B \sim \mathcal{D}$
3:   **(A) Quasimetric Learning** : *Section 4.2*
4:   **Compute Distance Losses** for each sample $i$ :
5:   $\mathcal{L}_{\text{close}}(s_i, s_i', \theta)$ *(Eq. 3)* : Calibrates one-step transitions
6:   $\mathcal{L}_{\text{push}}(s_i, g, \theta)$ *(Eq. 4)* : Pushes any state and goal apart
7:   **Update Quasimetric Parameters** $\theta$ :
8:   $\theta \leftarrow \theta - \eta \nabla_\theta \frac{1}{B} \sum_{i=1}^B (\mathcal{L}_{\text{close}} + \mathcal{L}_{\text{push}})$
9:   **Update Lagrange multiplier** $\lambda$ :
10:   $\lambda \leftarrow \lambda + \eta \nabla_\lambda \mathcal{L}_{\text{close}}$ : Adapt the penalty weight
11:   **(B) Policy Learning** : *Section 4.4*
12:   **Update Target Parameters** : $\mathcal{L}_T(s_i, s_i', w_i, \psi^-)$ *(Eq. 6)*
13:   Advantage Weighted Regression on the subgoal for $\pi_{\psi^-}$
14:   **Update Main Parameters** : $\mathcal{L}_M(s_i, w_i, g_i, \psi)$ *(Eq. 8)*
15:   Subgoal Advantage-Weighted Bootstrapping for $\pi_\psi$
16: **end for**
17: **(C) Keypoints Selection** : *Section 4.3*
18: **Update Keypoints** $\{z_k\}_{k=1}^K$ by solving a maximal
19: $\delta$-separated set problem on $\mathcal{D}$, using the learned
20: quasimetric $d_\theta$ and the separation distance $\delta$
21: **Return :** $d_\theta$ , $\pi_\psi$ , $\{z_k\}_{k=1}^K$

---

## 4.2. Geometric Foundation : Quasimetric Learning

To enable graph construction and policy learning, we need a calibrated quasimetric function $d_\theta(s, g)$ that estimates the minimum steps to reach $g$ from $s$. To model such a function, we implement **Interval Quasimetric Embeddings (IQE)** (Wang & Isola, 2022), within the **Quasimetric RL (QRL)** distance function learning framework (Wang et al., 2023).

IQE sets the distance as the aggregated length of intervals in a learned latent space. Given an encoder function $f_{\theta'}$ : $\mathcal{S} \to \mathbb{R}^{N \times M}$, the distance from $x \in \mathcal{S}$ to $y \in \mathcal{S}$ is formed by components that capture the total Lebesgue measure of unions of several intervals on the real line :

$$d_{\theta'}^{(i)}(x, y) = \left| \bigcup_{j=1}^M \left[ f_{\theta'}(x)_{ij}, \max \left( f_{\theta'}(x)_{ij}, f_{\theta'}(y)_{ij} \right) \right] \right| \quad (1)$$

Using the *maxmean* reduction (Pitis et al., 2020), we obtain IQE-*maxmean* with a single extra trainable parameter $\alpha \in [0, 1]$. Considering $\theta = [\theta', \alpha]$ and $\beta = 1 - \alpha$ we get :

$$d_\theta(x, y) = \alpha \cdot \max \left( d_{\theta'}^{(i)}(x, y) \right) + \beta \cdot \text{mean}\left( d_{\theta'}^{(i)}(x, y) \right) \quad (2)$$

**Theorem 4.1.** *IQE General and Universal Approximation (See Wang & Isola 2022) Consider any quasimetric space $(\mathcal{X}, d)$ where $\mathcal{X}$ is compact and $d$ is continuous. $\forall \epsilon > 0, \exists N$, there exist parameters $\theta = [\theta', \alpha]$ such that :*

$$\forall x, y \in \mathcal{X}, \ \left| d_\theta(x, y) - d(x, y) \right| \leq \epsilon .$$

By design, IQE satisfies all the quasimetric properties (Wilson, 1931), while QRL calibrates it by enforcing *local consistency* ($\mathcal{L}_{close}$) and *global separation* ($\mathcal{L}_{push}$) :

$$\mathcal{L}_{close}(\theta) = \mathbb{E}_{(s,s') \sim \mathcal{D}} \left[ \lambda \cdot \left( \texttt{relu}(d_\theta(s, s') - 1)^2 - \epsilon^2 \right) \right] \quad (3)$$

$$\mathcal{L}_{push}(\theta) = - \mathbb{E}_{(s,g) \sim \mathcal{D}} \left[ \omega \big( d_\theta(s, g) \big) \right] \quad (4)$$

where $\epsilon$ relaxes the local constraint and consequently eases the training, while $\omega$ is a monotonically increasing convex function to accelerate and to stabilize the learning process.

**Theorem 4.2.** *QRL Value Function Approximation (See Wang et al. 2023) Consider a compact state space $\mathcal{S}$ and an optimal undiscounted value function $V^*$. Supposing that the state space equals the goal space, if $\{d_\theta\}_{\theta \in \Theta}$ are universal approximators of quasimetric functions over $\mathcal{S}$, then $\forall \epsilon > 0, \exists \theta^* \in \Theta, \forall s, g \in \mathcal{S}$ :*

$$\mathbb{P}\Big[ \big| d_{\theta^*}(s, g) + (1+\epsilon)V^*(s, g) \big| \leq \sqrt{\epsilon} \Big] = 1 - \mathcal{O}\big( -\sqrt{\epsilon} \cdot \mathbb{E}[V^*] \big)$$

*i.e. the optimal distance function recovers $-V^*$ up to a scale, with probability of the order $1 - \mathcal{O}\big( -\sqrt{\epsilon} \cdot \mathbb{E}[V^*] \big)$ .*

Crucially, to prevent the optimization from ignoring small one-step constraints in favor of the global push, QRL use an adaptive Lagrange multiplier $\lambda$. This parameter is learned via gradient ascent to maximize the penalty on $\mathcal{L}_{\text{close}}$, ensuring the metric remains calibrated near 1 for transitions while maximizing separation between any states and goals.

## 4.3. Topological Abstraction : Keypoints Selection

We formalize state space mapping as extracting a heuristic to the Maximal $\delta$-Separated Set (Du & Wan, 2012) on the induced manifold. Formally, given a separation radius $\delta > 0$, we seek a finite set of keypoints $\mathcal{Z} \subset \mathcal{D}$ such that : (*separation*) they are all at least $\delta$-distant from each other ; (*covering*) every state inside $\mathcal{D}$ is within a $\delta$-distance of $\mathcal{Z}$.

**Iterative Wavefront Expansion.** Building a graph using random sampling or farthest point sampling can result in disconnected components or non-uniform covering. We propose *Iterative Wavefront Expansion* (IWE) as a greedy heuristic to the maximal $\delta$-separated set problem. Defining $d_{\mathrm{sym}}(x, y) := \max(d_\theta(x, y), d_\theta(y, x))$, we add the **closest** candidate $s \in \mathcal{D}$ to the set $\mathcal{Z}$ that satisfies $d_{\mathrm{sym}}(s, z) \geq \delta$, $\forall z \in \mathcal{Z}$ to enforce **separation** and promote **connectivity**.

This greedy strategy, detailed in *Algorithm 2*, grows the graph outwards like a wavefront, so that the keypoints added are reachable from the existing structure, preventing gaps while enforcing a near-uniform sparsity controlled by $\delta > 0$.

---

**Algorithm 2 AQM Iterative Wavefront Expansion**

---

**Require:** *Candidates $S = \mathcal{D}$ ; Quasimetric $d_\theta$ ; Repel radius $\delta$ .*
**Init:** $z_0 \sim S$ ; $\mathcal{Z} \leftarrow \{z_0\}$ ; $S \leftarrow \{ s \in S \mid d_{\mathrm{sym}}(s, z_0) \geq \delta \}$ .
1: **while** $|S| > 0$ **do**
2:     $z_{\mathrm{new}} \leftarrow \arg\min_{s \in S} \left( \min_{z \in \mathcal{Z}} d_{\mathrm{sym}}(s, z) \right)$
3:     $\mathcal{Z} \leftarrow \mathcal{Z} \cup \{z_{\mathrm{new}}\}$
4:     $S \leftarrow \{ s \in S \mid d_{\mathrm{sym}}(s, z_{\mathrm{new}}) \geq \delta \}$
5: **end while**
6: **Return** $\mathcal{Z}$ .

---

**Dataset Covering.** Assuming convergence to the optimal quasimetric $d^*$, and $d^*_{\mathrm{sym}}(x, y) := \max(d^*(x, y), d^*(y, x))$, *Algorithm 2* returns a $\delta$-cover of the dataset under $d^*_{\mathrm{sym}}$ .

**Proposition 4.3.** *Covering. Let $\mathcal{M}$ be a compact, path-connected subset of the state space. Assuming $\mathcal{D}$ is $\epsilon$-dense in $\mathcal{M}$ (i.e., $\forall x \in \mathcal{M}, \exists s \in \mathcal{D}, d^*_{sym}(x, s) < \epsilon$), the set of keypoints $\mathcal{Z}$ generated by Algorithm 2 is a $(\delta + \epsilon)$-cover of $\mathcal{M}$ : $\forall x \in \mathcal{M}, \min_{z \in \mathcal{Z}} d^*_{sym}(x, z) < \delta + \epsilon$ .*

AQM assumes sufficient coverage of the relevant support. This is often realistic in telemetry-rich settings (e.g., video games), but may be limiting in data-scarce applications. See *Appendices A* and *B* for more discussions and illustrations.

**Graph Construction.** From the set of keypoints selected, $\mathcal{Z} = \{z_1, \ldots, z_K\}$, we build a *directed* $G = (\mathcal{Z}, \mathcal{E})$. Given a radius $\tau > 0$, we connect $z_i, z_j \in \mathcal{Z}$ whenever $d^*_{\mathrm{sym}}(z_i, z_j) < \tau$, and add *both* directed edges $(i \to j)$ and $(j \to i)$, with their respective *time-to-reach* as edge weights.

**Proposition 4.4.** *Graph Connectivity. Following the above process, with $\tau = 2(\delta + \epsilon)$, the underlying undirected graph is connected (hence $G = (\mathcal{Z}, \mathcal{E})$ is strongly connected).*

### 4.4. Learning a Robust Policy

To navigate between keypoints, we train a goal-conditioned policy $\pi_\psi(a|s, g)$. Instead of training a single controller, we leverage a **Policy Bootstrapping** learning framework. While our learned quasimetric $d_\theta$ provides an unbounded and geometrically consistent signal superior to discounted value functions, directly extracting control actions from such metrics may be brittle as it would require model-based look-ahead (Wang et al., 2023). To avoid this, we leverage a Policy Bootstrapping framework (Zhou & Kao, 2025).

Instead of relying on a separate intermediate controller, this approach focuses on mastering atomic local behaviors, and repurposing them to reach the next subgoal. We identify and stitch together the specific local actions that contribute to reaching designated targets, ensuring robust edge traversal. During training we maintain a **target policy** $\pi_{\psi^-}$ and a **main policy** $\pi_\psi$. The target policy $\pi_{\psi^-}$ is trained via Quasimetric Advantage Weighted Regression (Q-AWR) to master *micro-behaviors*, to reach subgoals $w$ sampled from a near future.

$$\omega_{\mathrm{T}}(s, s', w) = \exp\left( \beta_{\mathrm{T}} \cdot \left( d_\theta(s, w) - d_\theta(s', w) \right) \right) \quad (5)$$

$$\mathcal{L}_{\mathrm{T}}(\psi^-) = -\mathbb{E}\left[ \omega_{\mathrm{T}}(s, s', w) \cdot \log \pi_{\psi^-}(a|s, w) \right] \quad (6)$$

Here $(s, a, s')$ is a transition. The main policy $\pi_\psi$ is trained to reach any goals $g$ (such as the keypoints) by distilling the target policy's skills. To ensure the short-term action towards $w$ actually contributes to the long-term goal $g$, we weigh the distillation loss leveraging Quasimetric Subgoal Advantage Weighted (Q-SAW), derived from AWR :

$$\omega_{\mathrm{M}}(s, w, g) = \exp\left( \beta_{\mathrm{M}} \cdot \left( d_\theta(s, g) - d_\theta(w, g) \right) \right) \quad (7)$$

$$\mathcal{L}_M(\psi) = \mathbb{E}\left[ \omega_{\mathrm{M}}(s, w, g) \cdot D_{\mathrm{KL}}\left( \pi_\psi(\cdot|s, g) \| \pi_{\psi^-}(\cdot|s, w) \right) \right] \quad (8)$$

Intuitively, if the subgoal $w$ lies on the optimal path to $g$ a high importance weight forces the $\pi_\psi$ to mimic the specific micro-behavior that leads to $w$, effectively stitching short but effective trajectories into long-horizon plans.

### 4.5. Geometric Consistency for Adaptive Planning

A novelty introduced by AQM is to exploit the learned quasimetric to enable principled test-time adaptation. For a neighboring pair $(z_i, z_{i+1})$, we set a time-to-reach budget $B(z_i, z_{i+1}) := (1 + \kappa) \cdot d_\theta(z_i, z_{i+1})$, as an upper bound of traversal time, with $\kappa \geq 0$ a fixed control-slack parameter.

Inference relies on the shortest path $\mathcal{P} = (s, z_1, \ldots, z_n, g)$ over the topological graph, computed via the Floyd-Warshall algorithm described in *Section 3*. To handle environmental shifts at test-time (such as a blocked path), we monitor the accumulation of the elapsed time as the agent traverses an edge. If this value exceeds the predicted time-to-reach budget, the learned agent considers the edge as unreliable, marks it as untraversable $\left( d(z_i, z_{i+1}) = \infty \right)$ in the graph, replan an alternative route, enabling zero-shot replanning.

## 5. Experiments

To empirically validate AQM, we conduct a comprehensive evaluation across a diverse suite of goal-conditioned tasks. Our experiments span standard benchmarks from OGBench (Park et al., 2025a) as well as environments designed to test zero-shot replanning capabilities from Continual NavBench (Kobanda et al., 2025). We assess performance against flat, hierarchical and graph-based Offline RL methods. Reported results represent the average success rates across multiple seeds, distinguishing between standard inference and those requiring adaptation at test time. Implementation details and configurations are provided in *Appendices B*, *C*, and *D*.

*Table 1.* **Evaluating Performance** (%) **on Navigation Tasks from Offline Data.** Baseline results are reported from Baek et al. (2025). Results are averaged over 4 seeds (50 rollouts per goal). Bold values indicate performance with 95% reach of the top performing method.

| Dataset Type | Dataset | GCBC | GCIQL | QRL | CRL | HILP | HIQL | SAW | GAS | AQM |
|---|---|---|---|---|---|---|---|---|---|---|
| Locomotion | antmaze-medium-navigate | $33 \pm 6$ | $75 \pm 5$ | $82 \pm 8$ | $\mathbf{95} \pm \mathbf{1}$ | $\mathbf{96} \pm \mathbf{1}$ | $\mathbf{95} \pm \mathbf{1}$ | $\mathbf{97} \pm \mathbf{1}$ | $\mathbf{96} \pm \mathbf{1}$ | $\mathbf{98} \pm \mathbf{1}$ |
| | antmaze-large-navigate | $24 \pm 3$ | $33 \pm 5$ | $75 \pm 4$ | $86 \pm 5$ | $87 \pm 4$ | $\mathbf{90} \pm \mathbf{2}$ | $\mathbf{90} \pm \mathbf{2}$ | $\mathbf{93} \pm \mathbf{1}$ | $\mathbf{94} \pm \mathbf{2}$ |
| | antmaze-giant-navigate | $0 \pm 0$ | $0 \pm 0$ | $14 \pm 4$ | $15 \pm 6$ | $53 \pm 3$ | $67 \pm 6$ | $70 \pm 4$ | $78 \pm 3$ | $\mathbf{88} \pm \mathbf{2}$ |
| Stitching | antmaze-medium-stitch | $43 \pm 8$ | $27 \pm 7$ | $67 \pm 11$ | $57 \pm 8$ | $\mathbf{96} \pm \mathbf{1}$ | $92 \pm 3$ | $95 \pm 2$ | $\mathbf{98} \pm \mathbf{1}$ | $\mathbf{98} \pm \mathbf{2}$ |
| | antmaze-large-stitch | $2 \pm 4$ | $10 \pm 3$ | $20 \pm 2$ | $14 \pm 6$ | $34 \pm 3$ | $72 \pm 5$ | $64 \pm 9$ | $\mathbf{97} \pm \mathbf{1}$ | $\mathbf{95} \pm \mathbf{2}$ |
| | antmaze-giant-stitch | $0 \pm 0$ | $0 \pm 0$ | $0 \pm 0$ | $0 \pm 0$ | $0 \pm 0$ | $1 \pm 1$ | $1 \pm 2$ | $\mathbf{88} \pm \mathbf{4}$ | $\mathbf{90} \pm \mathbf{3}$ |
| Exploratory | antmaze-medium-explore | $3 \pm 3$ | $12 \pm 1$ | $1 \pm 1$ | $1 \pm 2$ | $40 \pm 7$ | $32 \pm 3$ | $27 \pm 7$ | $\mathbf{98} \pm \mathbf{0}$ | $\mathbf{99} \pm \mathbf{1}$ |
| | antmaze-large-explore | $0 \pm 0$ | $1 \pm 1$ | $0 \pm 1$ | $0 \pm 0$ | $2 \pm 2$ | $3 \pm 4$ | $5 \pm 4$ | $\mathbf{94} \pm \mathbf{3}$ | $\mathbf{94} \pm \mathbf{5}$ |

## 5.1. Experimental Setup

### 5.1.1. ENVIRONMENTS

We evaluate navigation methods on the OGBench AntMaze suite (*Figure 3*), across different temporal horizons and data distributions. To assess test-time adaptation, we employ Continual NavBench (*Figure 4*), a modular video-game-like benchmark with multiple configurations, where agents must distinguish between *jumpable* low platforms and impassable high walls. A brief overview of reinforcement learning and navigation in video games is available in *Appendix A*.

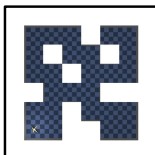 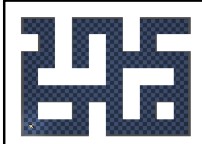 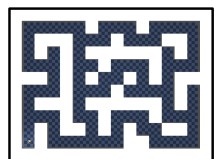

| (a) **Medium** | (b) **Large** | (c) **Giant** |
|---|---|---|

*Figure 3.* **OGBench Environments :** AntMaze Configurations.

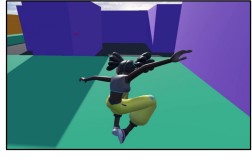 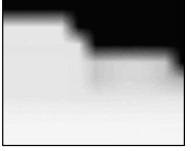 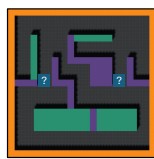

| (a) **Human Playing** | (b) **Raycast Vision** | (c) **AmazeVille** |
|---|---|---|

*Figure 4.* **Continual NavBench :** Visualization & Configurations.

### 5.1.2. BASELINES

We compare AQM against a representative suite of offline GCRL baselines : flat methods **GCBC** (Ding et al., 2019), **GCIQL** (Kostrikov et al., 2022), **QRL** (Wang et al., 2023), **CRL** (Eysenbach et al., 2022), **SAW** (Zhou & Kao, 2025); hierarchical ones **HIQL** (Park et al., 2023); and graph-based ones **HILP** (Park et al., 2024b) and **GAS** (Baek et al., 2025).

## 5.2. Main Results : How well does AQM perform ?

### 5.2.1. HOW EFFECTIVE IS AQM ON NAVIGATION TASKS ?

As shown in *Table 1*, our proposed framework, **AQM**, **achieves state-of-the-art performance** on the OGBench AntMaze suite, consistently outperforming or equaling flat, hierarchical, and graph-based baselines on all environments.

Standard flat methods (GCIQL, QRL, and CRL) perform reasonably well on `medium` tasks but suffer a catastrophic collapse in performance on `large` or `giant` navigation tasks, notably when using `stitching` datasets (achieving $0 - 15\%$ success rates). While hierarchical methods like HIQL and the SAW distillation-based methods provide a significant improvement by reasoning over subgoals, they still exhibit a degradation as the temporal horizon increases. For instance, in `antmaze-giant-navigate`, they only reach a success rate about 70 %, significantly falling behind AQM and GAS. These results suggests that even hierarchical reasoning struggle with longer horizon offline environments.

The results clearly indicate that graph-based frameworks (such as AQM and GAS) are the only class of navigation algorithms capable of maintaining high success rates in longer horizon settings (above 78 % on `giant` tasks). By explicitly modeling the topology of the environments, these methods effectively *stitch* together sparse trajectories that flat policies cannot connect, while significantly achieving better performances than hierarchical methods. Beyond this structural advantage, AQM sets a new ceiling for the benchmark, achieving 88 % and 90 % on the most difficult navigation and stitching tasks of the considered benchmark.

### 5.2.2. HOW IS THE SUCCESS-COMPACTNESS TRADE-OFF ?

The comparative results displayed in *Table 2* highlight a critical distinction in how AQM and GAS methods leverage the learned environment's topology. While both are graph-based, **GAS exhibits a heavy reliance on graph density to maintain performance**. As the number of nodes in GAS decreases, its success rate degrades sharply. For instance, in `antmaze-giant-navigate`, going from 4286 to 268 nodes results in a 10 % return loss (74 % → 64 %), whereas **AQM remains highly effective with orders of magnitude fewer nodes**, achieving 88 % return with only 158 nodes.

This disparity in keypoint count represents a decisive barrier for practical deployment. Algorithmic complexity for path-finding methods, like Floyd-Warshall, scale polynomially with the count ($O(|\mathcal{Z}|^3)$)). Moreover, assigning the current state to the nearest keypoint requires querying all keypoints. Consequently, as the horizon gets longer, graphs with dense keypoint coverage scales poorly and cause a heavy burden.

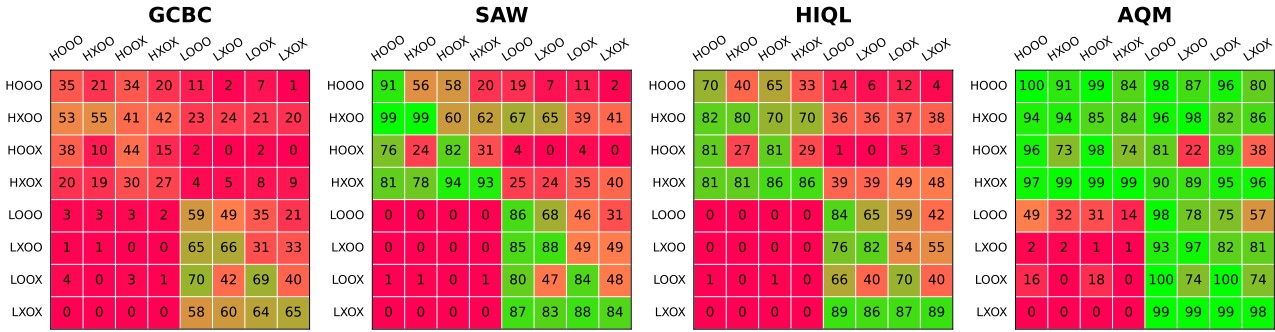

*Figure 5.* **Evaluating Test-time Adaptation.** Performances are presented as heatmaps where rows represent the training environment and columns represent the evaluation environment. The diagonal values represent standard evaluation setup (training and evaluation environments match), while off-diagonal entries quantify test-time replanning capabilities under specific environmental shifts.

In resource-constrained applications such as modern video games, where computational budget is strictly partitioned, an overhead is prohibitive. Deploying a method that requires 15000 nodes (e.g., GAS on `antmaze-large-explore`) is a prohibitive burden, prone to starving other critical processes. By achieving state-of-the-art results with a *leaner* graph, AQM ensures a computationally feasible reasoning in production environments where AI tools must coexist with other demanding real-time systems. In addition, Appendix F provides a more detailed comparison of AQM and GAS.

*Table 2.* **Comparison of AQM and GAS Methods.**

| Dataset | Method | # Nodes | Success (%) |
|---|---|---|---|
| antmaze-giant-navigate | GAS | 4286 | 74 ± 3 |
| | GAS | 978 | 77 ± 3 |
| | GAS | 431 | 71 ± 2 |
| | GAS | 268 | 64 ± 2 |
| | AQM | **158** | **88** ± **2** |
| | AQM | **109** | **86** ± **2** |
| | AQM | **67** | **83** ± **1** |
| antmaze-giant-stitch | GAS | 12901 | 81 ± 3 |
| | GAS | 1966 | 88 ± 4 |
| | GAS | 688 | 80 ± 4 |
| | GAS | 375 | 69 ± 2 |
| | AQM | **163** | **90** ± **3** |
| | AQM | **113** | **88** ± **3** |
| | AQM | 71 | 82 ± 1 |
| antmaze-large-explore | GAS | 15143 | 98 ± 1 |
| | GAS | 2499 | 94 ± 3 |
| | GAS | 1126 | 88 ± 8 |
| | GAS | 679 | 90 ± 4 |
| | AQM | **177** | **91** ± **7** |
| | AQM | **114** | **94** ± **5** |
| | AQM | **68** | **88** ± **6** |

### 5.2.3. TO WHAT EXTENT DOES OUR APPROACH HANDLE TEST-TIME ADAPTATION IN CHANGING ENVIRONMENTS ?

We evaluate test-time adaptation on Continual NavBench, where mazes vary by gate status (**O** : Open / **X** : Closed) and obstacles (**H** : high or *non-jumpable* / **L** : low or *jumpable*).

We compare a subset of representative baselines per family of approaches : **GCBC** (as a default method), **SAW** (as the best-performing flat policy), **HIQL** (as the state-of-the-art hierarchical method), and **AQM** (as a the best trade-off graph-based method : with high-performance and low cost).

In *Figure 5*, the diagonal results represent evaluations where training and testing environments match. The best results are achieved by AQM (avg. 98), followed by SAW (avg. 88), while HIQL (avg. 80) lags further behind likely due to compounding errors between high-level subgoal generation and low-level execution. GCBC (avg. 53) is struggling on all maps, possibly due to long-horizon and mode averaging. These differences are even more pronounced on the $L$-maps.

The other results reveal the adaptation capabilities. **Within the same obstacles category, AQM demonstrates robust replanning** : as a previously open gate is closed ($O \rightarrow X$), the agent must find an alternative path to reach a goal. **AQM successfully performs zero-shot replanning, using the edge reliability detection, rerouting in the pruned graph.** Moreover, agents trained on *non-jumpable* obstacles maps transfer effectively to *jumpable* ones. This is because the *walk-around* behaviors remain valid, although sub-optimal.

However, transferring from $L$-maps to $H$-maps exposes an essential limitation of test-time adaptation : **data support**. In $L$-map datasets, optimal trajectories involve jumping over obstacles. When deployed in $H$-maps, these actions are rendered useless, and the learned agents lack the specific *long-detour* skills required. The graph cannot stitch a valid solution from the available transitions. While AQM aims at maximizing the utility of existing knowledge, adaptation remains strictly bounded by the behavioral coverage of the training data. When the necessary recovery skills are absent from the available data, test-time replanning is very difficult.

## 5.3. Ablation Study : What Makes AQM Work ?

### 5.3.1. ON DIFFERENT KEYPOINTS SELECTION STRATEGIES

We evaluate Iterative Wavefront Expansion (*Section 4.3*) (IWE) against Random Sampling (RND) and Farthest Point Sampling (FPS). For a fair comparison, FPS and IWE share the same repel radius $\delta$, while RND has various sampling budgets. Resulting mappings are visualized in *Appendix H*.

*Table 3.* **Study of Keypoint Selection Strategies.**

| Dataset | Selection Strategy | # Nodes | Success (%) |
|---|---|---|---|
| antmaze-medium-navigate | RND | 25 | 98 ± 1 |
| | RND | **50** | **96 ± 1** |
| | RND | 100 | 96 ± 2 |
| | FPS | 116 | 97 ± 1 |
| | IWE | **47** | **98 ± 1** |
| antmaze-large-navigate | RND | 25 | 78 ± 7 |
| | RND | 50 | 90 ± 6 |
| | RND | 100 | 94 ± 2 |
| | FPS | 150 | 93 ± 3 |
| | IWE | **71** | **94 ± 2** |
| antmaze-giant-navigate | RND | 25 | 50 ± 10 |
| | RND | 50 | 54 ± 7 |
| | RND | 100 | 60 ± 5 |
| | FPS | 200 | 72 ± 10 |
| | IWE | **158** | **88 ± 2** |

As shown in *Table 3*, IWE is effective and efficient. While FPS yields respectable performance, it results in higher node counts. This inefficiency likely arises since FPS greedily selects distant points, thus it tends to sample misevaluated outliers due to the QRL *push* loss *(Eq. 4)*. In contrast, RND tends to fails on larger environment due to uneven coverage.

This confirms that an adapted structured selection process is essential to maintain high-level planning reliability in AQM.

### 5.3.2. ON THE INFLUENCE OF THE REPEL RADIUS

The repel radius $\delta$ (see *Section 4.3*), mitigates the trade-off between efficient mapping and navigation performance.

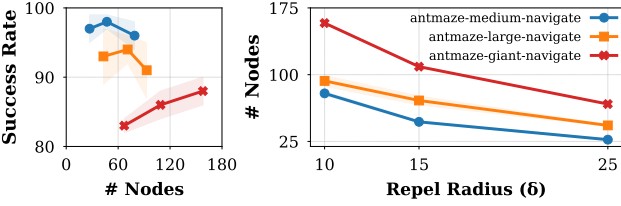

*Figure 6.* Performance Trade-offs Relatively to the Repel Radius.

As shown in *Figure 6*, increasing $\delta$ sharply reduces node counts across all tasks. While success remains stable on `medium` and `large` mazes, in the `giant` one it degrades as the graph thins. This indicates that the graph becomes sparser, success rate is sensitive to the density of subgoals.

*Figure 7* illustrates how our keypoints selection strategy provides a uniform coverage. While a small radius yields a dense mapping, larger ones filter out local redundancies, providing a more parsimonious topological representation.

These results suggest that for a reasonable $\delta$, performance is bounded by the agent's ability to reach specified subgoals.

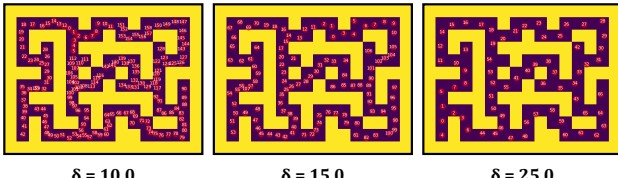

| δ = 10.0 | δ = 15.0 | δ = 25.0 |
|---|---|---|

*Figure 7.* Repel Radius Influence on the `giant` Maze Mapping.

### 5.3.3. ON THE LOW POLICY ACTION CONTROL

We study the impact of different low-level policy on the performance of AQM. We compare Behavior Cloning (BC), Quasimetric Advantage Weighted Regression (Q-AWR), and Quasimetric Subgoal Advantage Weighted (Q-SAW) which we use by default in AQM. Explanations on each of these methods are provided in *Section 4.4*.

As shown in *Table 4*, Q-AWR provides marginal gains over BC but fail on exploratory tasks, while Q-SAW remains efficient. Thus, not leveraging the hierarchical nature of GCRL (Q-AWR) is likely insufficient for long-horizon tasks. Instead, it seems better to leverage hierarchical distillation when using quasimetrics (Q-SAW) to learn useful behaviors.

*Table 4.* **Study of Low Policy Strategies.**

| Dataset | Low Policy | # Nodes | Success |
|---|---|---|---|
| antmaze-giant-navigate | BC | 152 | 21 ± 4 |
| | Q-AWR | 152 | 25 ± 5 |
| | Q-SAW | 158 | **88 ± 2** |
| antmaze-giant-stitch | BC | 162 | 84 ± 7 |
| | Q-AWR | 162 | 83 ± 7 |
| | Q-SAW | 163 | **90 ± 3** |
| antmaze-large-explore | BC | 112 | 0 ± 0 |
| | Q-AWR | 112 | 0 ± 0 |
| | Q-SAW | 114 | **94 ± 5** |

## 6. Discussion

We introduced **Adaptive Quasimetric Mapping (AQM)**, a learning framework bridging the gap between geometric representation learning and efficient long-horizon planning. By leveraging covering theory for graph construction, **AQM** approximates a $\delta$-covering of the dataset support. This parsimonious structure enables state-of-the-art performance on navigation benchmarks, computational efficiency, while allowing test-time adaptation. Comprehensive evaluations and analysis show that **AQM** offers a robust and principled solution for deployment in resource-constrained settings.

Nonetheless, adaptive capabilities are bounded by the data support : if specific recovery behaviors are absent, test-time adaptation is rendered difficult and we are likely to require retraining. Furthermore, our focus was on deterministic settings. Future work could extend **AQM** to stochastic environments notably via probabilistic graph formulations.

## Impact Statement

This paper presents work whose goal is to advance the field of machine learning. There are few potential societal consequences of our work, none of which we feel must be specifically highlighted here. We do not claim readiness for real-world safety-critical deployment; our experiments are limited to standard Offline RL benchmarks in simulation.

## Acknowledgements

This project was provided with computer resources by GENCI at IDRIS thanks to the grant 2024-A0181016109 on the supercomputer Jean Zay's CSL partition.

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

# Appendices

This supplementary material complements the main paper with additional results, theoretical details, extended discussion, and implementation specificities. Its purpose is to provide a deeper insight, a practical grounding, and support reproducibility.

List of the **Appendices** :

- *Appendix A* – **Video Games, Reinforcement Learning, and Navigation**

  Provides contextual background on offline reinforcement learning in modern video game environments. It discusses the nature of large-scale gameplay datasets, practical constraints in game production, and motivates the relevance of goal-conditioned navigation and adaptive planning in this domain.

- *Appendix B* – **Experimental Setup and Environment Details**

  Details the environments and datasets used in the experiments, including OGBench AntMaze configurations and Continual NavBench. This appendix complements *Section 5.1.1* by clarifying environment choices and task variations.

- *Appendix C* – **Baselines**

  Presents the baseline algorithms we used for comparison in our experiments, introduced in *Section 5.1.2*, categorizing them into flat (**GCBC**, **GCIQL**, **CRL**, **QRL**, **SAW**), hierarchical (**HIQL**), and graph-based (**HILP**, **GAS**) approaches.

- *Appendix D* – **Implementation Details**

  Provides the implementation details, including network architectures, hyperparameter settings, used for benchmarking.

- *Appendix E* – **Proofs**

  Contains the formal mathematical demonstrations for the theoretical claims made in *Section 4.2* and *Section 4.3*

- *Appendix F* – **Comparison of AQM and GAS**

  Contains detailed explanations about how AQM and GAS are similar, how they differ, and why their differences matter.

- *Appendix G* – **Additional Results**

  Presents additional quantitative results and ablation studies regarding **AQM**, extending the analysis from *Section 5*.

- *Appendix H* – **Learned State-Space Mappings**

  Offers a qualitative analysis of the topological graphs learned by our approach. Through visualizations of the generated keypoints on different layouts, it illustrates the impact of the repel radius $\delta$ and on the resulting graph structure.

# A. Video Games, Reinforcement Learning, and Navigation

## A.1. Virtual Worlds, Players, and Data

Modern video games are persistent, high-fidelity simulations that track every interactive entity in real time, from character coordinates to environmental triggers (Lucas et al., 2013; Zoeller, 2013). Worldwide platforms routinely generate petabytes of play-logs per month through automatic telemetry pipelines. These logs, often archived for matchmaking or analytics, form unusually clean, richly annotated datasets that are orders of magnitude larger than typical robotics corpora, making video games the ideal frontier for Offline Reinforcement Learning (RL) (Levine et al., 2020; Macaluso et al., 2024).

However developing video games comes with various production constraints: state spaces are often partially observable (Kurniawati, 2022), and both computational and commercial budgets require hard guarantees on players good experience and safety (Kobanda et al., 2023; Gu et al., 2024). Thus, learning frameworks not only have to be data-efficient but also controllable and adaptable to the frequent content updates that characterize the *Games as a Service* model (Clark, 2014).

## A.2. From Scripted NPCs to Adaptive Agents

Traditional Non-Player Characters (NPCs) rely on finite-state machines or hand-authored behavior trees (Zhu, 2019). While robust, these scripts are brittle: a slight change in map geometry or a new game mechanic often requires manual re-scripting. While Deep Reinforcement Learning agents have mastered complex video games (Shao et al., 2019), with titles like *Dota 2* (Berner et al., 2019) through massive self-play, the computational cost of such training is prohibitive for most studios.

Consequently, interest has shifted toward *offline* or *offline-to-online* methods that can learn from existing player replays. Our proposed framework, **AQM**, bridges the gap between the robustness of classical *nav-meshes* (Cui & Shi, 2011) and the flexibility of Deep Reinforcement Learning by modeling an environment topology directly from pre-collected player traces.

## A.3. Practical Challenges in Game Environments

Goal-Conditioned RL (GCRL) is a natural fit for modern games (Alonso et al., 2020; Balloni et al., 2023), in particular Offline GCRL : every quest, checkpoint, or capture flag supplies an explicit goal signal, while replay archives furnish millions of feasible, human-level trajectories on which to train models.

Nevertheless, three practical challenges still limit adoption :

**Coverage & Directionality :** Replay buffers often sample only a thin manifold of the reachable world. Furthermore, game mechanics are often one-way (e.g., ledge drops, teleporters). AQM addresses this by leveraging a quasimetric distance that respects the inherent asymmetry of these transitions.

**Stochasticity :** Human behavior and random event triggers (e.g., loot drops or portal destinations) break the deterministic assumptions of standard planners ; consequently policies must be robust to such distributional change, which is not guaranteed by most frameworks. This is a current limitation of AQM and most methods it is tested against.

**Environmental Shifts:** This is the most critical hurdle. If a developer moves a wall or adds a platform, most offline policies are rendered useless. AQM solves this by considering the learned graph as a dynamic object that can be updated at test-time.

## A.4. Adaptive Planning and Environment Design

AQM (see *Section 4*) operationalizes the concept of a *data-driven nav-mesh* designed for zero-shot adaptation. By monitoring a *time-to-reach budget* between topological keypoints, the agent detects obstructions, a common occurrence in the modular layouts of our Continual NavBench (Kobanda et al., 2025) experiments, and instantly reroutes through the graph. This mimics human spatial reasoning, where a player finding a blocked path intuitively bypasses it using an internal map of the world without needing to relearn basic motor skills or retrain a policy.

However, the effectiveness of this replanning is fundamentally a function of dataset coverage. The agent can only reroute if the offline data contains the alternative behaviors required for the new path it must follows. While the massive telemetry available to game companies usually ensures sufficient behavioral coverage, classical benchmarks often lack the specific diversity, such as localized backtracking, needed for recovery in tight corridors. In such *out-of-distribution* cases where a feasible path simply does not exist in the prior data, replanning remains impossible and would necessitate retraining.

Continual NavBench environments are specifically designed to highlight these dynamics and failure cases. Their modular initialization system adds or removes blocks and platforms to simulate *level patches*, purposefully breaking previous optimal trajectories to force the agent to demonstrate the adaptive robustness, that is the hallmark of the AQM learning framework.

# B. Environment Details

## B.1. Agents and Mazes

### B.1.1. OGBENCH

**OGBench** (Park et al., 2025a) is a standardized suite of offline goal-conditioned learning tasks. It includes AntMaze, a navigation environment in which a ant-like agent must reach arbitrary goals within bounded mazes of various horizon length.

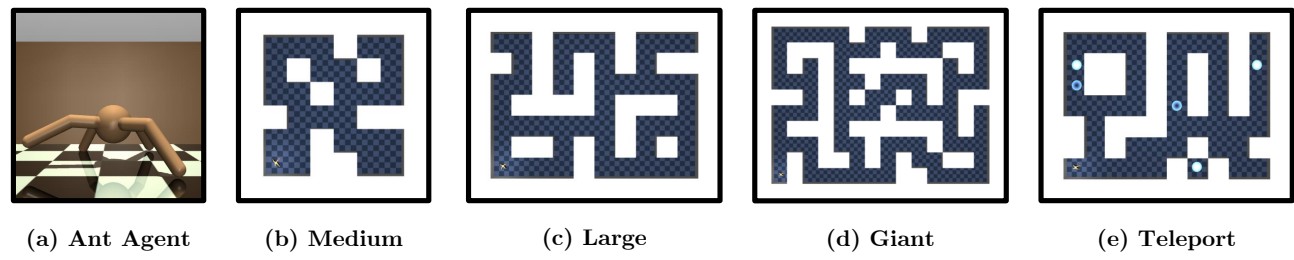

(a) Ant Agent      (b) Medium      (c) Large      (d) Giant      (e) Teleport

Figure 8. **OGBench** : AntMaze Agent & Configurations.

Four maze variants appear in OGBench (see *Figure 8*) : `medium` with short trajectory lengths testing basic navigation ; `large` with a longer horizon, increasing the decision complexity ; `giant` scales the maze again, yielding an even larger layout. Although we focused on deterministic environment, we mention the `teleport` configuration which has the same size as the `large` with stochastic portals that randomly chose exits (potentially dead ends).

### B.1.2. CONTINUAL NAVBENCH

While AntMaze environment allows to efficiently assess navigation performance, it lacks features needed to fully understand how topographic variations affect an agent. Therefore we leverage a video-game like 3D navigation benchmark, namely **Continual NavBench** (Kobanda et al., 2025), that offer diverse mazes with more explainable spatial challenges.

We use the **AmazeVille** suite, with $60\ m \times 60\ m$ mazes. They have a finite set of start and goal positions, with two subsets of maps : some with high blocks (H), *i.e.* not jumpable obstacles ; others with low blocks (L), *i.e.* jumpable ones.

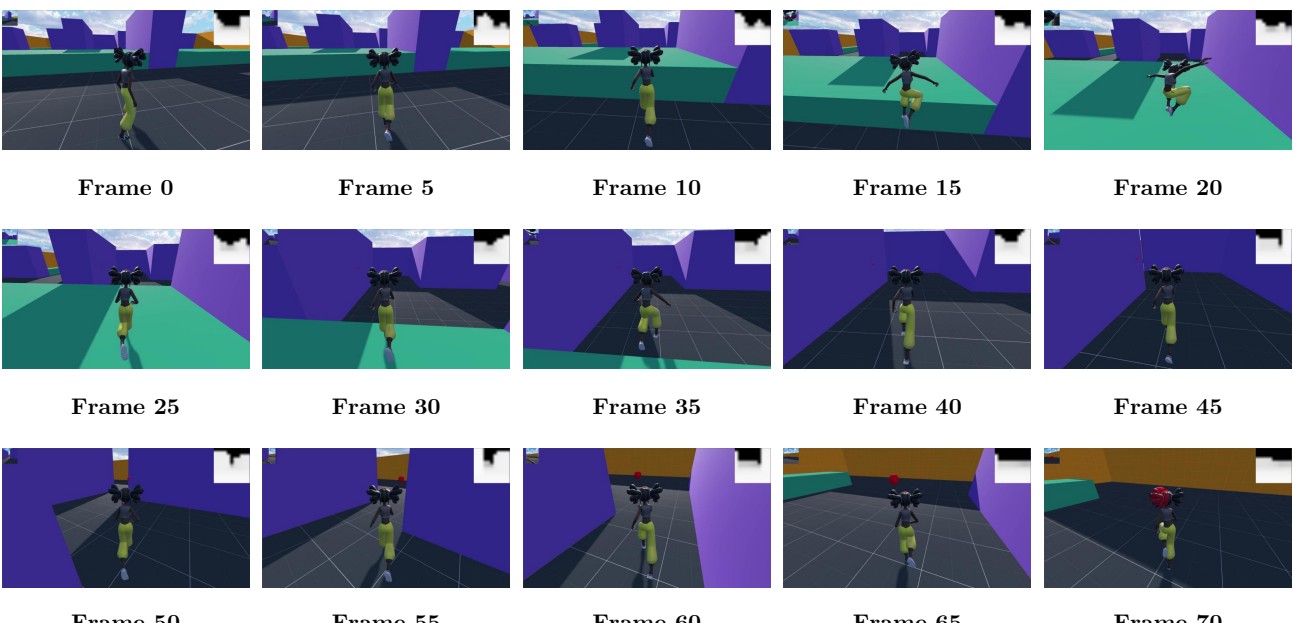

Frame 0      Frame 5      Frame 10      Frame 15      Frame 20

Frame 25      Frame 30      Frame 35      Frame 40      Frame 45

Frame 50      Frame 55      Frame 60      Frame 65      Frame 70

Figure 9. **Continual NavBench** : Visualization of a Human Playing on A-LOOO.

*Figure 9* shows a trajectory generated by a human player on one of the maze **A-LOOO**. The naming indicate we are using whether specific doors are open (**O**) or not (**X**), and if movable green blocks are in high positions (**H**) or low positions (**L**), providing a clear way to distinguish between different maze configurations. *Figure 10* displays the different configurations.

Although the maximum number of features an observation may have is 12440, we use $(x, y, z)$ coordinates as the goal space (which provided by default) and the current position, the agent orientation (angle in radian according to the vertical axis), its velocity (in meters per second), and the depth image (as $11 \times 11$ raycasts) from its position to the visible nearest obstacles.

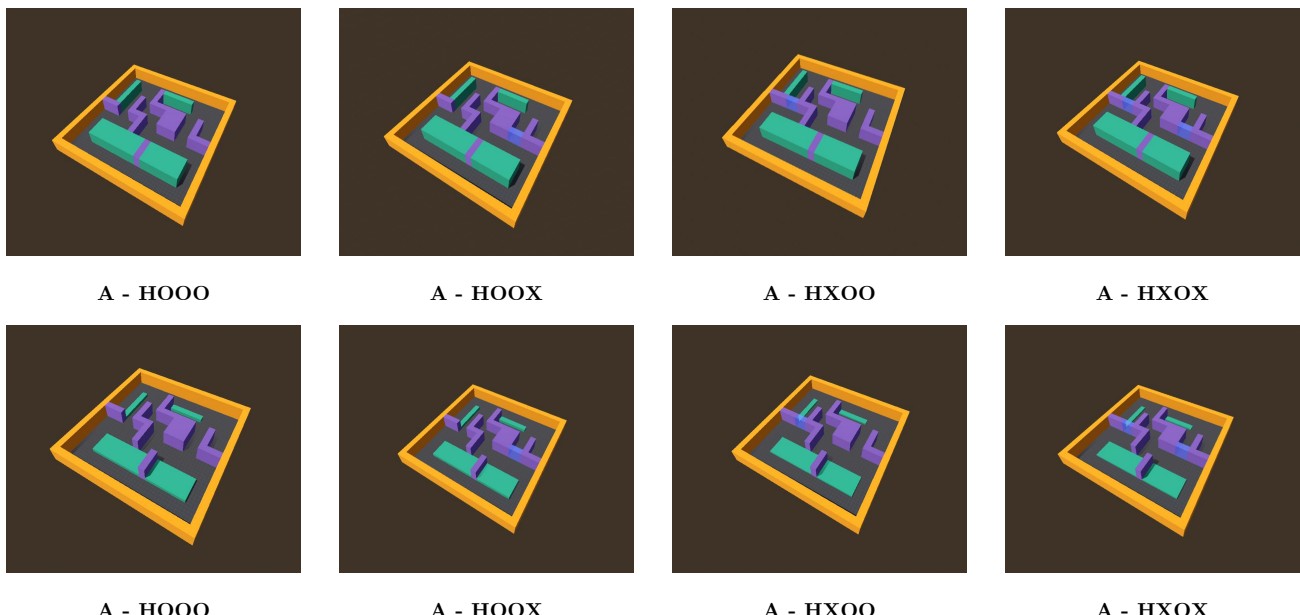

| A - HOOO | A - HOOX | A - HXOO | A - HXOX |

| A - HOOO | A - HOOX | A - HXOO | A - HXOX |

*Figure 10.* **Continual NavBench** : AmazeVille Maze Configurations.

## B.2. Datasets

### B.2.1. OGBENCH

Each AntMaze configurations provide multiple offline datasets : `navigate` contains expert rollouts (1000000 transitions) where a noisy expert wanders toward different goals ; `stitch` breaks those trajectories into shorter ones (200 steps each), forcing to *stitch* discontiguous trajectories ; `explore` consists of random exploratory trajectories (5000000 transitions), designed to test whether the agent can learn navigation skills from extremely low-quality (yet high-coverage)

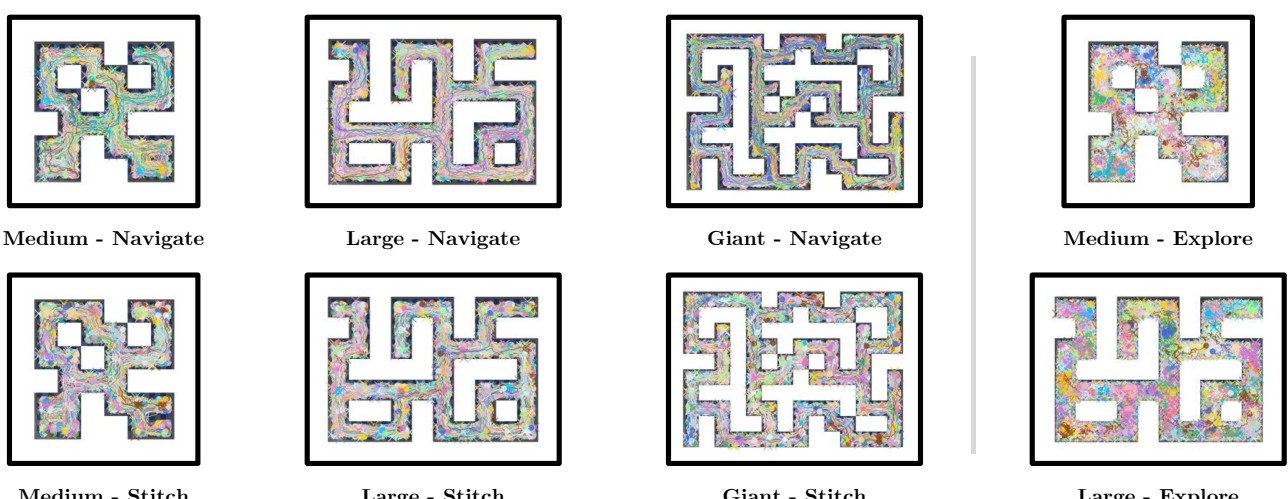

| Medium - Navigate | Large - Navigate | Giant - Navigate | Medium - Explore |

| Medium - Stitch | Large - Stitch | Giant - Stitch | Large - Explore |

*Figure 11.* **OGBench** : AntMaze Trajectories Visualization.

*Figure 11* displays all available trajectories for the different dataset types. We notably see how densely the trajectories cover each maze configuration for `navigate`, `stitch`, and even more the `explore` datasets. Although we don't investigate the subject, as it is not the main focus of our research, we hypothesis that `explore` datasets are more informative to efficiently learn a distance with QRL (Wang et al., 2023), as they provide a broader covering of the set of possible transitions.

B.2.2. CONTINUAL NAVBENCH

AmazeVille datasets are all human generated, through hours of play sessions, with 100 episodes for each maze configuration. Consequently, these datasets capture strategies that are potentially valuable for training bots to exhibit human-like behavior.

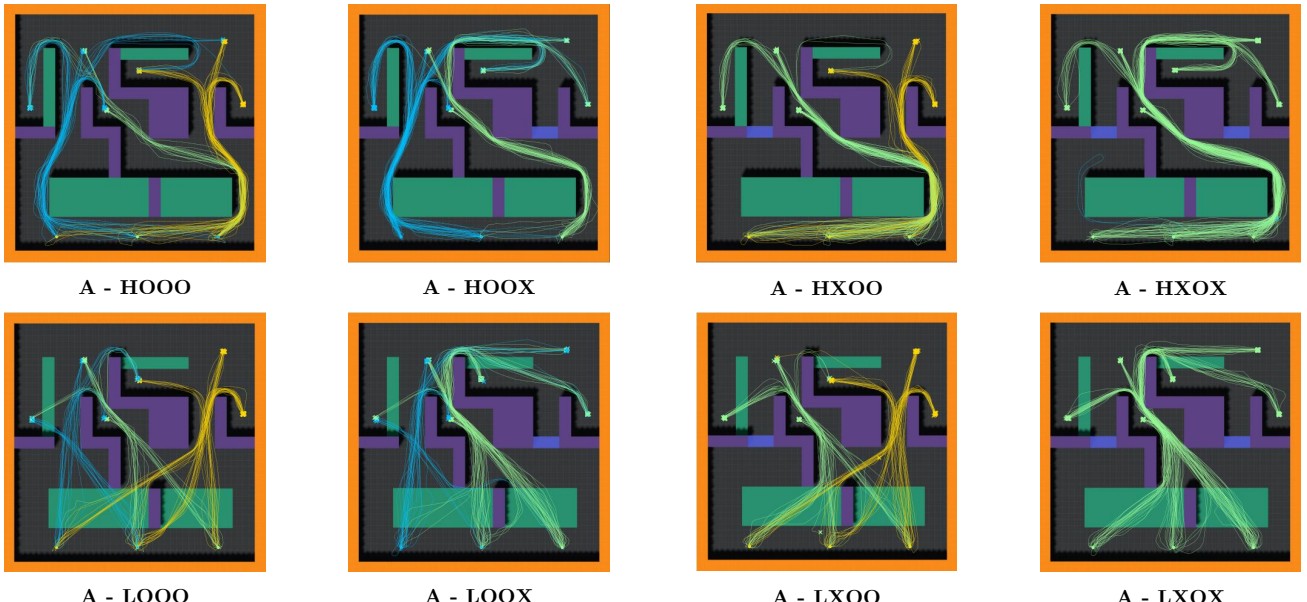

Figure 12. **Continual NavBench** : AmazeVille Trajectories Visualization.

# C. Baselines

This section details the baseline methods used to evaluate AQM. We categorize these into three architectural paradigms (Liu et al., 2022) : **flat methods**, which map state-goal pairs directly to actions ; **hierarchical methods**, which decompose tasks using latent subgoals; and **graph-based methods**, which learn and levergae topological structures for navigation.

## C.1. Flat Methods

**Goal-Conditioned Behavioral Cloning (GCBC)** (Ding et al., 2019) is the simplest baseline. It treats the problem as a supervised learning task, where a policy is trained to map state-goal pairs directly to the actions found in the dataset. While robust and easy to implement, **GCBC** is inherently limited by the quality of the behavioral data : it cannot *stitch* trajectories or improve upon the suboptimal demonstrations. **AQM** differs by employing an RL-based distance function that allows the agent to reason about reachability and optimality beyond simple imitation.

**Goal-Conditioned Implicit Q-Learning (GCIQL)** (Kostrikov et al., 2022; Park et al., 2025a) extends Implicit Q-Learning to the goal-conditioned setting. It avoids the overestimation issue common in offline RL by using expectile regression to estimate the value of the best action within the dataset's support without ever querying out-of-distribution actions. While **GCIQL** is a strong baseline for medium-horizon tasks, it struggles with the compounding value-estimation errors of long-horizon navigation. **AQM** addresses this by decomposing the problem into a sequence of local, graph-guided sub-tasks.

**Contrastive Reinforcement Learning (CRL)** (Eysenbach et al., 2022) leverages contrastive learning to represent the value function. It trains a critic to distinguish between states that are near each other in time and states that are far, effectively learning a representation where the dot product of state-goal embeddings correlates with reachability. However, **CRL** distances are often uncalibrated and represent relative similarities rather than true temporal distances. **AQM** improves upon this by using a quasimetric formulation specifically calibrated to estimate time-to-reach values.

**Quasimetric Reinforcement Learning (QRL)** (Wang et al., 2023) is a principled approach that learn quasimetric. It enforces the triangle inequality as a structural prior during training, allowing for robust distance estimation. While **AQM** adopts a similar quasimetric learning framework, we expand it by integrating a parsimonious keypoint selection strategy and an adaptive planning mechanism, moving from a pure distance-learning method to a space mapping and navigation system.

**Subgoal Advantage-Weighted Policy Bootstrapping (SAW)** (Zhou & Kao, 2025) is a flat offline GCRL method that targets long-horizon control without hierarchical inference. It trains a single goal-conditioned policy by sampling intermediate *subgoals* from dataset trajectories and regressing the policy toward the actions that reach these subgoals, with updates *advantage-weighted* so that subgoals inducing stronger progress toward the final goal dominate learning. This provides a simple, effective form of stitching while staying within dataset support. **AQM** leverages a related subgoal decomposition idea, but with a learned *time-to-reach quasimetric* as the core decision signal, deriving advantages and graph edges from it.

### C.2. Hierarchical Methods

Hierarchical methods leverage a multi-level structure where a high-level policy selects intermediate subgoals for a low-level controller to achieve. To represent this paradigm, we compare against the current state-of-the-art.

**Hierarchical Implicit Q-Learning (HIQL)** (Park et al., 2023) is the SOTA representative for Hierarchical GCRL. it uses a high-level policy to pick latent subgoals from the dataset, which a low-level policy then pursues. This hierarchy allows the agent to reason across longer temporal horizons than flat methods. However, **HIQL**'s reliance on latent spaces makes the policy brittle to environmental changes not seen during training. **AQM** provides greater robustness by using an explicit keypoint graph that can be updated dynamically to handle blocked paths or layout shifts.

### C.3. Graph-Based Methods

Graph-based methods explicitly build a topological map of the environment, where nodes represent states and edges represent feasible transitions. We compare our approach against the most recent approaches in this category.

**Hilbert Foundation Policies (HILP)** (Park et al., 2024b) learn a geometric state embedding $\phi$ into a Hilbert space where Euclidean distances $\|\phi(s) - \phi(g)\|$ approximate long-horizon temporal distances, and train a latent-conditioned policy to follow *directions* in this learned geometry. Goal-reaching can then be performed by choosing directions consistent with the embedding, using $\phi$ as an implicit topological abstraction. However, HILP's success hinges on the accuracy of the learned geometry and it does not directly provide a mechanism to invalidate broken connections under test-time topological changes. **AQM** instead builds an explicit sparse graph and prunes edges online via a calibrated time-to-reach budget.

**Graph-Assisted Stitching (GAS)** (Baek et al., 2025) is the SOTA representative for graph-based navigation. It builds a graph by clustering the dataset and creating edges based on temporal proximity. While **GAS** is highly effective at solving complex mazes, it reflects a broader trend in the field where SOTA performance is increasingly attainable, yet few methods remain geometrically principled or computationally parsimonious. **GAS** typically requires a high density of nodes and lacks a mechanism for test-time adaptation. **AQM** distinguishes itself by utilizing a mathematically grounded dominating set selection strategy to build sparser graphs and by introducing a scalable time-to-reach budget system for test-time adaptation.

## D. Implementation Details

We use **JAX** (Bradbury et al., 2018) as our learning framework. Unless otherwise stated, we follow the experimental protocol from **OGBench** (Park et al., 2025a) and keep all unspecified hyperparameters identical ; the deviations are reported below.

### D.1. Network Architectures

**Backbones.** If not otherwise stated, all networks (quasimetric encoder and policy networks) are MLPs with 3 hidden layers of width 512 and GELU activations. Only for IQE we use LayerNorm in each MLP block (after the nonlinearity).

**Interval Quasimetric Embedding (IQE).** We parameterize the quasimetric using Interval Quasimetric Embeddings (IQE) (Wang & Isola, 2022) with 512 components of size 1 (i.e., a $512 \times 1$ latent reshaping). In the original IQE formulation, a common choice is 64 components of size 8. We found the two parameterizations to yield comparable performance in our setting, but the $512 \times 1$ variant is significantly faster in our JAX implementation due to simpler element-wise operations.

**Actor Policies.** Policies are goal-conditioned and follow the same MLP backbone described above. Unless otherwise stated, we keep action parameterization, squashing, and initialization identical to the **OGBench** defaults.

## D.2. Hyperparameters & Optimization

We use the Adam optimizer for all modules. All policy networks are trained with learning rate $\eta_\pi = 3 \times 10^{-4}$ and batch size $B = 1024$. The IQE quasimetric encoder is trained with learning rate $\eta_d = 1 \times 10^{-4}$, the Quasimetric RL Lagrange multiplier is initialized at $\lambda = 1.0$ and trained with a learning rate $\eta_\lambda = 1 \times 10^{-2}$. For advantage-weighted policy updates (from SAW and HIQL, to Q-AWR and Q-SAW objectives) we set the temperature to $\alpha = 3.0$. All of our experiments run for $10^6$ gradient steps. Results are averaged over the same **four seeds** per training environment and dataset.

Training is performed on NVIDIA V100 GPUs. Typical wall-clock time ranges from about 2 hours on **OGBench** to about 4 hours on **Continual NavBench**, largely due to evaluation overhead (simulation runtime) rather than gradient computation.

In QRL (for $\mathcal{L}_{\text{push}}$ in Eq. 4), we use the strictly monotonically increasing convex function, with $m$ as offset and $s$ as scale :

$$\omega(x) \triangleq - \operatorname{softplus}\big(m - x, \beta = s\big) = - \frac{\operatorname{softplus}\big(s \cdot (m - x)\big)}{s}.$$

*Table 5.* Offset and Scale Hyperparameters per Dataset.

| Dataset | Offset ($m$) | Scale ($s$) |
|---|---|---|
| antmaze-medium-{navigate, stitch, explore} | 50.0 | 0.01 |
| antmaze-large-{navigate, stitch, explore} | 125.0 | 0.01 |
| antmaze-giant-{navigate, stitch, explore} | 250.0 | 0.01 |
| AmazeVille | 100.0 | 0.01 |

**Keypoint Separation Sweep.** We sweep the IWE radius $\delta$ and report the best performance for each dataset. For AntMaze we sweep $\delta \in \{10, 15, 25\}$, and for AmazeVille we sweep $\delta \in \{5, 10\}$. For low-level subgoal horizons, we sweep $\{5, 10, 15\}$.

## D.3. Data Sampling & Processing

We use the **OGBench** offline datasets and the **Continual NavBench** datasets described in *Appendix B*.

**Goal and subgoal sampling.** In AQM, goals are sampled from future states along the same trajectory, except for `explore` datasets where goals are sampled uniformly at random. Subgoals are sampled at a fixed future offset (see previous section).

**Direction-Conditioned Policies (Continual NavBench).** On Continual NavBench, we condition both low-level and high-level policies on the *direction* to the (sub)goal, i.e., $\Delta g = \frac{g - s}{\|g - s\|}$, instead of the absolute goal coordinate. We apply this to all goal-conditioned methods with hierarchical interfaces (SAW, HIQL, AQM).

**Scope Note (Continual NavBench).** Continual NavBench is designed to stress adaptation under distribution shift; its logged trajectories may not provide diverse alternative paths. Since the goal space in this setting lies in $\mathbb{R}^3$, with additional engineering (e.g., injecting stochasticity, memory, or explicit exploration during data collection), policy-only baselines might partially mitigate some failure modes, but would likely lack AQM's test-time graph pruning mechanism and lack behind.

## D.4. Discussions

**On Evaluation Protocols.** For AntMaze baselines, we report the numbers from the GAS (Baek et al., 2025) and evaluate our methods under the *same* protocol (dataset versions, evaluation procedure, and reporting). This reuse is standard practice on established benchmarks and allows rapid, controlled comparisons without redundant retraining when protocols are aligned. The only baseline we reran for AntMaze is SAW (Zhou & Kao, 2025), since it is not reported in the GAS paper; we ran SAW using the official implementation and default settings, matching the same evaluation protocol used for our methods.

**On Runtime Comparisons.** We do not report cross-paper wall-clock comparisons because implementations span different frameworks, engineering choices, and heterogeneous hardware stacks ; *naïve* timing would mix algorithmic effects with software-level artifacts. A fair runtime study would require a unified, carefully engineered setup, which was out of scope here ; we therefore focus on protocol-matched performance metrics and on the algorithmic analysis presented in the paper.

# E. Proofs

In this appendix, we provide the demonstrations for the theoretical claims made in *Section 4.2* and *Section 4.3*.

We focus on two properties of the keypoint set produced by the **Iterative Wavefront Expansion** (**IWE**) algorithm :

(i) Covering guarantee ( *Proposition 4.3* ) ;

(ii) Connectivity of the induced keypoint graph ( *Proposition 4.4* ) .

## E.1. Assumptions and Notation

Let the state space be a subset $\mathcal{X} \subset \mathbb{R}^n$, $\mathcal{D} \subseteq \mathcal{X}$ be a (finite) set of states, and let $\mathcal{D} \subseteq \mathcal{M} \subseteq \mathcal{X}$ denote its support.

**A1. Compactness and Path-Connectedness.** We assume $\mathcal{M}$ is compact and a path-connected subset of $\mathcal{X}$.

**A2. Optimal Quasimetric.** We assume convergence to an optimal time-to-reach quasimetric $d^* : \mathcal{X} \times \mathcal{X} \to \mathbb{R}^+$.

We define the symmetrized distance as : $d^*_{\text{sym}}(x, y) := \max \left\{ d^*(x, y) , d^*_{\text{sym}}(y, x) \right\}$.

**A3. Dataset Density.** Let $\epsilon > 0$. We assume $\mathcal{D}$ is $\epsilon$-dense in $\mathcal{M}$ w.r.t. $d^*_{\text{sym}} : \forall x \in \mathcal{M}$ , $\exists s \in \mathcal{D}$ s.t. $d^*_{\text{sym}}(x, s) < \epsilon$ .

## E.2. Preliminary Lemmas

**Lemma E.1** (**Symmetrized Metric**). *Assuming A2, $d^*_{sym}$ is a metric on $\mathcal{X}$.*

*Proof.* We verify the metric axioms :

**Non-negativity.** Since $d^*(x, y) \geq 0$ and $d^*(y, x) \geq 0$, we have $d^*_{\text{sym}}(x, y) = \max\{d^*(x, y), d^*(y, x)\} \geq 0$ .

**Identity of indiscernibles.** If $x = y$, then $d^*(x, y) = d^*(y, x) = 0$, hence $d^*_{\text{sym}}(x, y) = 0$.
Conversely, if $d^*_{\text{sym}}(x, y) = 0$, then $d^*(x, y) = 0$ and $d^*(y, x) = 0$, which implies $x = y$.

**Symmetry.** By definition : $d^*_{\text{sym}}(y, x) = \max\{d^*(y, x), d^*(x, y)\} = d^*_{\text{sym}}(x, y)$.

**Triangle inequality.** By the directed triangle inequality for $d^*$ :

$$d^*(x, z) \leq d^*(x, y) + d^*(y, z) \leq d^*_{\text{sym}}(x, y) + d^*_{\text{sym}}(y, z).$$

Applying the same argument to the reversed direction yields :

$$d^*(z, x) \leq d^*(z, y) + d^*(y, x) \leq d^*_{\text{sym}}(y, z) + d^*_{\text{sym}}(x, y) = d^*_{\text{sym}}(x, y) + d^*_{\text{sym}}(y, z).$$

Taking the maximum of the two left-hand sides gives :

$$d^*_{\text{sym}}(x, z) = \max\{d^*(x, z), d^*(z, x)\} \leq d^*_{\text{sym}}(x, y) + d^*_{\text{sym}}(y, z),$$

which proves the triangle inequality, and the lemma. □

**Lemma E.2** (*Algorithm 2* **Terminates**). *Assuming A2,* Algorithm 2 *terminates in a finite amount of iterations, at most $|\mathcal{D}|$ .*

*Proof.* At each iteration, the algorithm selects a keypoint $z \in S$ and updates : $S \leftarrow \left\{ s \in S \mid d^*_{\text{sym}}(s, z) \geq \delta \right\}$.

Since $d^*_{\text{sym}}(z, z) = 0 < \delta$, the selected point $z$ is always removed by this update, i.e., $z \notin S$ after the pruning step.

As long as the condition $S \neq \emptyset$ holds, the cardinality strictly decreases : $|S_{k+1}| \leq |S_k| - 1$ . Since $\mathcal{D}$ is finite and $S_0 = \mathcal{D}$, this decrease cannot occur indefinitely. Hence after at most $|\mathcal{D}|$ iterations, the algorithm reach $S = \emptyset$ and terminates. □

## E.3. Dataset covering

**Dataset Covering.** Assuming **A2**, *Algorithm 2* returns a $\delta$-cover of the dataset under $d_{\text{sym}}^*$ .

*Proof. Algorithm 2* initializes $S \leftarrow \mathcal{D}$ and iteratively selects a keypoint $z \in S$, adds it to $\mathcal{Z}$, then removes from $S$ all points $s$ such that $d_{\text{sym}}^*(s, z) < \delta$. The algorithm terminates only when $S = \emptyset$. Therefore every dataset point $s \in \mathcal{D}$ must have been removed at some iteration due to being within $d_{\text{sym}}^*$-distance $< \delta$ of a selected keypoint, i.e.: $\forall s \in \mathcal{D}, \exists z \in \mathcal{Z}$ s.t. $d_{\text{sym}}^*(s, z) < \delta$. Hence $\mathcal{Z}$ is a $\delta$-cover of $\mathcal{D}$ under $d_{\text{sym}}^*$ . $\qquad\square$

## E.4. Proof of *Proposition 4.3* (Covering)

*Proposition 4.3.* *Assuming **A1**, **A2**, and **A3**, the set of keypoints $\mathcal{Z}$ generated by* Algorithm 2 *is a $(\delta + \epsilon)$-cover of $\mathcal{M}$.*

*Proof.* Fix any $x \in \mathcal{M}$. By $\epsilon$-density, $\exists s \in \mathcal{D}$ such that $d_{\text{sym}}^*(x, s) < \epsilon$ .

By the Dataset Covering result above, $\exists z \in \mathcal{Z}$ such that $d_{\text{sym}}^*(s, z) < \delta$ . Using *Lemma E.1* :

$$d_{\text{sym}}^*(x, z^\star) \leq d_{\text{sym}}^*(x, s) + d_{\text{sym}}^*(s^\star, z^\star) < \epsilon + \delta.$$

Since $x$ was arbitrary, $\mathcal{Z}$ is a $(\delta + \epsilon)$-cover of $\mathcal{M}$ . $\qquad\square$

## E.5. Graph Construction

**Graph Construction.** Assuming **A1**, **A2**, and **A3**, with the selected keypoints $\mathcal{Z} = \{z_1, \ldots, z_K\}$ by *Algorithm 2*, we build a directed graph $G = (\mathcal{Z}, \mathcal{E})$. Given a radius $\tau > 0$, we connect $z_i, z_j \in \mathcal{Z}$ whenever $d_{\text{sym}}^*(z_i, z_j) < \tau$, and add *both* directed edges $(i \rightarrow j)$ and $(j \rightarrow i)$ with weights $d^*(z_i, z_j)$ and $d^*(z_j, z_i)$, respectively.

## E.6. Proof of *Proposition 4.4* (Connectivity)

**Proposition 4.4.** *Assuming **A1**, **A2**, and **A3**, and following the aforementioned construction process, with $\tau = 2(\delta + \epsilon)$, the underlying undirected graph is connected (hence $G$ is strongly connected) .*

*Proof.* Let $R := \delta + \epsilon$ and $\tau := 2R$.

Assume for contradiction that the underlying undirected graph is disconnected.

Then $\mathcal{Z}$ can be partitioned into two non-empty disjoint sets $A$ and $B$ such that no edge crosses the cut.

By the edge rule, this implies that $\forall\, a \in A, \forall\, b \in B\,:\, d_{\text{sym}}^*(a, b) \geq \tau = 2R$ .

We define two open subsets of $\mathcal{M}$ (as unions of open balls) :

$$U_A := \bigcup_{a \in A} \{x \in \mathcal{M} : d_{\text{sym}}^*(x, a) < R\} \quad,\quad U_B := \bigcup_{b \in B} \{x \in \mathcal{M} : d_{\text{sym}}^*(x, b) < R\}.$$

**Covering.** By Proposition 4.3, $\mathcal{Z} = A \cup B$ is an $R$-cover of $\mathcal{M}$, hence $\mathcal{M} \subseteq U_A \cup U_B$ .

**Disjointedness.** If $x \in U_A \cap U_B$, then $\exists\, a \in A,\, \exists\, b \in B$ such that $d_{\text{sym}}^*(x, a) < R$ and $d_{\text{sym}}^*(x, b) < R$. By *Lemma E.1*, $d_{\text{sym}}^*(a, b) \leq d_{\text{sym}}^*(a, x) + d_{\text{sym}}^*(x, b) < 2R = \tau$, contradicting $d_{\text{sym}}^*(a, b) \geq \tau$. Hence $U_A \cap U_B = \emptyset$.

**Contradiction.** We have written $\mathcal{M}$ as a union of two disjoint non-empty open sets, contradicting **A1**.

Therefore the underlying undirected graph must be connected.

**Strong connectivity of $G$.** Since for every undirected adjacency we add both directed edges, any undirected path in the underlying graph lifts to a directed path in $G$. Thus $G$ is strongly connected. $\qquad\square$

### E.7. Discussion: assumptions vs. practice

The previous results are stated under clean assumptions to make the guarantees explicit.

We now clarify :

(i) Why these assumptions are reasonable in our experimental setting ;

(ii) When they may fail ;

(iii) How the theory connects to the practical implementation .

**On A1 (Compactness and Path-Connectedness).**

We assume $\mathcal{M} \subseteq \mathcal{X} \subset \mathbb{R}^n$ is compact and path-connected. In the navigation domains considered here, the relevant state variables are either inherently bounded (e.g., positions in a bounded map, bounded joint angles or velocities after clipping, bounded observations by design), or are effectively restricted to a bounded region of interest by the dataset support. This motivates compactness as a modeling assumption.

Path-connectedness is also natural for the *reachable* subset of states induced by a single environment layout : for instance, a continuous workspace (or maze free-space) typically forms one connected component when obstacles do not fully separate the space. In practice, we observe this behavior on our datasets: see *Appendix B* for qualitative visualizations in OGBench showing that the collected states concentrate on a dense, coherent region.

*Failure modes.* A1 can be violated when the dataset mixes multiple disconnected supports : data aggregated from distinct maps with no overlap ; multiple rooms separated by closed doors never opened in data ; multi-modal resets that place the agent in disjoint regions with no trajectories connecting them. In such cases, IWE still produces a cover of $\mathcal{D}$, but *Proposition 4.4* may fail because the true support contains multiple connected components; correspondingly, a single connected planning graph is not the right abstraction. Our method is therefore most appropriate when the dataset covers a single connected region of the task support.

**On A2 (Convergence to the Optimal Quasimetric, and a Realistic Extension).**

The previous proofs assume convergence of the learned quasimetric to $d^*$ to keep the statements simple. In practice, we only have access to an approximation $d_\theta$. A natural extension is to introduce a uniform approximation error $\varepsilon_q$ such that, for all $x, y$ in the relevant support :

$$\left| d_\theta(x, y) - d^\star(x, y) \right| \leq \varepsilon_q \quad , \quad \left| d_{\text{sym},\theta}(x, y) - d^\star_{\text{sym}}(x, y) \right| \leq \varepsilon_q \, ,$$

which would propagate through the inequalities used in Proposition 4.3 and Proposition 4.4. Concretely, covering and connectivity radii would inflate by an additive term on the order of $\varepsilon_q$ (e.g., $(\delta + \epsilon)$ becomes $(\delta + \epsilon + \varepsilon_q)$, and $\tau$ inflates accordingly), making explicit how approximation error weakens the guarantees.

Quantifying the quality of a learned quasimetric in large continuous spaces is difficult: there is no single scalar metric that simultaneously captures triangle inequality tightness, calibration to time-to-reach, and usefulness for planning under distribution shift. In our work, we therefore rely on two practical diagnostics: (i) qualitative inspection of learned mappings (*Appendix H*), and (ii) downstream behavioral validation. In particular, consistent replanning success likely indicates that the induced distances are sufficiently informative in the regions used for graph construction and planning.

**On A3 (Dataset $\epsilon$-density).**

The $\epsilon$-density assumption is a standard way to formalize "coverage" of the relevant support. In telemetry-rich settings (e.g., video games), large-scale logging often produces dense state visitation in the regions of interest. This is consistent with our empirical setting: visualizations in *Appendix B* suggest that OGBench datasets are sufficiently dense for the covering and connectivity arguments to be meaningful.

*Limitation under scarce data.* When data are sparse, large holes appear in the support and $\epsilon$ becomes large, weakening the theoretical radius $\tau = 2(\delta + \epsilon)$ and, more importantly, degrading planning quality in practice. In that regime, AQM is expected to fail or require additional mechanisms (e.g., exploration, data augmentation, or stronger priors on dynamics) since sparse offline data fundamentally limits long-horizon navigation.

**Unknown $\epsilon$ and choice of $\tau$ (we do not set $\tau = 2(\delta + \epsilon)$).**

The parameter $\epsilon$ is not directly observable in high-dimensional continuous-control domains. Hence, we do not instantiate the theoretical threshold $\tau = 2(\delta + \epsilon)$. Instead, we use a simple conservative proxy consistent with the IWE intuition:

$$\tau := 2\delta \quad \text{(optionally with a small slack).}$$

This choice empirically yields well covering and connected graphs across our benchmarks, while avoiding density estimation.

**Batch-based Keypoint Selection (JAX-friendly).**

For compilation and memory reasons, we run IWE on a large subset $\mathcal{D}_B \subset \mathcal{D}$ rather than the full dataset. The theoretical statements then apply to the sampled support induced by $\mathcal{D}_B$. In practice, we mitigate this by setting $|\mathcal{D}_B|$ to 1024 and repeating IWE over multiple batches, selecting the one with the most keypoints. Empirically, we find that the resulting graphs remain stable once the batch is large enough to reflect the dataset geometry.

**(Optional) Connectivity by construction.**

A robust implementation-level alternative, independent of $\epsilon$, is to record at insertion time the attachment distance $\rho_{k+1} := \min_{z \in Z_k} d_{\mathrm{sym}}(z_{k+1}, z)$ and explicitly include the corresponding parent edge. This guarantees connectivity for any threshold $\tau \geq \max_k \rho_{k+1}$ and can be used as a safety mechanism, especially when density is hard to assess.

# F. Comparison of AQM and GAS

AQM and GAS share a system pattern : both combine a learned temporal geometry, graph-based planning, and a low-level goal-conditioned controller. The differences lie in the abstraction principle, the graph construction, and test-time adaptation.

*Table 6.* Comparison Between AQM and GAS. The purpose is to clarify overlap and design differences.

| Aspect | AQM | GAS | Implication |
|---|---|---|---|
| Backbone | Quasimetric + Graph Planning + Low-level Control | Temporal-distance Representation + Graph Planning + Low-level Control | The broad pipeline is shared ; GAS is the closest graph-based baseline. |
| Temporal geometry | Directed time-to-reach quasimetric learned with IQE/QRL | Temporal-distance Representation (TDR) with Euclidean latent distance | AQM is more aligned with directed reachability budgets ; but this is not isolated as the source of the node gap. |
| Graph construction | Greedy maximal $\delta$-separated sparse cover over dataset states | Temporal-efficiency filtering followed by TD-aware clustering | Sparse support cover versus filtered temporal-distance clustering. |
| Graph support | Keypoints are carefully selected states from the available dataset | Keypoints are cluster centroids built in the latent space | AQM remains directly support grounded, matching its cover analysis. |
| Theory | Covering and connectivity analysis under density and approximation assumptions | Graph construction is motivated and ablated empirically, but there are no theoretical guarantees | AQM makes the sparse abstraction layer theoretically explicit, whereas GAS focus on empirical results. |
| Low-level control | Ablations with different policies Q-SAW, BC and Q-AWR | DDPG+BC with TD-aware subgoal sampling | The controller determines whether a sparse graph can be exploited reliably. |
| Test-time adaptation | Budget-based edge invalidation and graph rerouting | No analogous topology-update rule is evaluated in the method | AQM adds support-bounded graph maintenance under topology change. |
| Empirical scope | Navigation, compactness, and topology-change adaptation | Broader task scope, including navigation and manipulation | GAS is broader ; AQM is narrower but studies adaptation and efficiency. |

The current experiments isolate two factors inside AQM. First, Table 3 shows that keypoint selection materially affects the compactness–success trade-off. Second, Table 4 shows that low-level control is critical for exploiting a sparse graph. What the current experiments do not isolate is a full cross-method causal decomposition between AQM and GAS. We therefore interpret Table 2 as evidence of a different operating point rather than proof that one component alone explains the full gap.

# G. Additional Results

## G.1. Navigation

*Table 7.* **Sensitivity to subgoal distance (H-Step) and separation radius $\delta$ on `antmaze-medium`.** Entries report success rate (%).

| Dataset Type | AntMaze Task | H-Step | $\delta = 10.0$ | $\delta = 15.0$ | $\delta = 25.0$ |
|---|---|---|---|---|---|
| Locomotion | navigate | 10 | $95.6 \pm 2.4$ | $97.6 \pm 2.3$ | $95.6 \pm 1.5$ |
| | navigate | 15 | $96.4 \pm 2.3$ | $95.4 \pm 4.3$ | $97.2 \pm 1.3$ |
| | navigate | 25 | $95.8 \pm 2.3$ | $98.0 \pm 1.4$ | $96.8 \pm 1.6$ |
| Stitching | stitch | 10 | $97.0 \pm 1.9$ | $93.8 \pm 8.4$ | $96.2 \pm 2.2$ |
| | stitch | 15 | $96.6 \pm 2.3$ | $91.8 \pm 9.0$ | $96.2 \pm 1.1$ |
| | stitch | 25 | $97.6 \pm 1.8$ | $96.2 \pm 0.8$ | $97.4 \pm 1.8$ |
| Exploratory | explore | 10 | $99.6 \pm 0.5$ | $99.4 \pm 0.9$ | $98.6 \pm 1.3$ |
| | explore | 15 | $99.4 \pm 0.9$ | $99.0 \pm 0.7$ | $98.6 \pm 1.3$ |
| | explore | 25 | $99.4 \pm 0.9$ | $97.8 \pm 1.8$ | $99.0 \pm 1.0$ |

*Table 8.* **Sensitivity to subgoal distance (H-Step) and separation radius $\delta$ on `antmaze-large`.** Entries report success rate (%).

| Dataset Type | AntMaze Task | H-Step | $\delta = 10.0$ | $\delta = 15.0$ | $\delta = 25.0$ |
|---|---|---|---|---|---|
| Locomotion | navigate | 10 | $91.8 \pm 1.6$ | $92.2 \pm 3.6$ | $92.2 \pm 2.6$ |
| | navigate | 15 | $90.8 \pm 3.5$ | $94.0 \pm 1.6$ | $93.4 \pm 4.1$ |
| | navigate | 25 | $92.2 \pm 1.6$ | $90.4 \pm 4.3$ | $91.2 \pm 4.1$ |
| Stitching | stitch | 10 | $94.8 \pm 1.9$ | $93.8 \pm 3.1$ | $87.8 \pm 8.0$ |
| | stitch | 15 | $92.4 \pm 4.0$ | $90.4 \pm 3.6$ | $90.6 \pm 2.2$ |
| | stitch | 25 | $92.8 \pm 4.0$ | $92.0 \pm 1.0$ | $89.4 \pm 10.1$ |
| Exploratory | explore | 10 | $90.6 \pm 7.3$ | $94.0 \pm 4.5$ | $83.2 \pm 25.7$ |
| | explore | 15 | $86.4 \pm 16.4$ | $87.8 \pm 5.9$ | $81.0 \pm 22.4$ |
| | explore | 25 | $86.2 \pm 2.8$ | $78.6 \pm 22.2$ | $88.2 \pm 5.7$ |

*Table 9.* **Sensitivity to subgoal distance (H-Step) and separation radius $\delta$ on `antmaze-giant`.** Entries report success rate (%).

| Dataset Type | AntMaze Task | H-Step | $\delta = 10.0$ | $\delta = 15.0$ | $\delta = 25.0$ |
|---|---|---|---|---|---|
| Locomotion | navigate | 10 | $85.2 \pm 7.0$ | $84.0 \pm 2.5$ | $80.6 \pm 3.2$ |
| | navigate | 15 | $87.8 \pm 1.5$ | $85.6 \pm 1.9$ | $82.8 \pm 1.3$ |
| | navigate | 25 | $85.0 \pm 4.8$ | $85.6 \pm 3.4$ | $84.2 \pm 2.6$ |
| Stitching | stitch | 10 | $86.4 \pm 3.8$ | $82.0 \pm 2.1$ | $81.0 \pm 9.1$ |
| | stitch | 15 | $90.4 \pm 3.3$ | $88.0 \pm 3.1$ | $82.2 \pm 1.1$ |
| | stitch | 25 | $90.4 \pm 4.2$ | $87.6 \pm 4.0$ | $80.0 \pm 4.8$ |

## G.2. Replanning

### G.2.1. REPLANNING – BENCHMARK

*Figure 13* shows the evaluation on a cross-environment generalization task: diagonal entries are in-distribution (training and evaluation environments match), while off-diagonal entries measure test-time adaptation under environment shifts. SAW and HIQL improve over GCBC and generalize to some shifted environments, but still exhibit large off-diagonal drops under harder shifts. In contrast, AQM maintains strong performance not only on the diagonal but also across many off-diagonal entries, indicating effective zero-shot replanning: when transitions become invalid under shifts, AQM can prune and reroute rather than committing to the original plan.

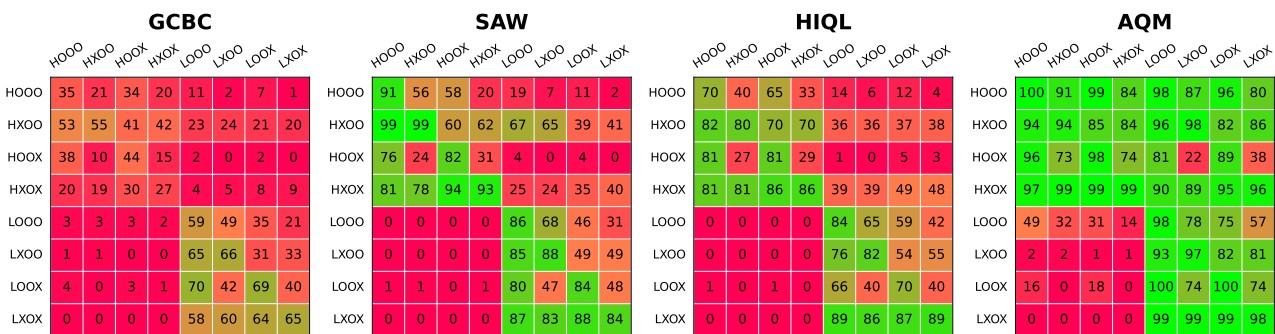

*Figure 13.* **Evaluating Test-time Adaptation.** Performances are presented as heatmaps where rows represent the training environment and columns represent the evaluation environment. The diagonal values represent standard evaluation setup (training and evaluation environments match), while off-diagonal entries quantify test-time replanning capabilities under specific environmental shifts.

### G.2.2. REPLANNING – ABLATION

*Figure 14* shows an ablation on two AQM design parameters: (i) the keypoint separation radius $\delta$, which controls how sparse the keypoint set is (smaller $\delta$ yields more keypoints) ; (ii) the subgoal sampling distance $K$, which controls how far ahead subgoals are sampled during training. Increasing $K$ from $5$ to $10$ generally improves transfer and adaptation (off-diagonal entries), with gains most visible in harder settings ($L$-maps $\rightarrow$ $H$-maps), consistent with longer-horizon subgoals improving the ability to bridge between keypoints. Varying $\delta$ induces a trade-off between coverage and robustness: smaller $\delta$ produces denser graphs that can perform well in-distribution but may be less stable under pruning, whereas larger $\delta$ yields sparser abstractions that remain robust provided the graph stays sufficiently connected. Overall, the most uniform performance across shifts (visually the "greenest" off-diagonal pattern) is obtained for the combination $\delta = 10$ and $K = 10$.

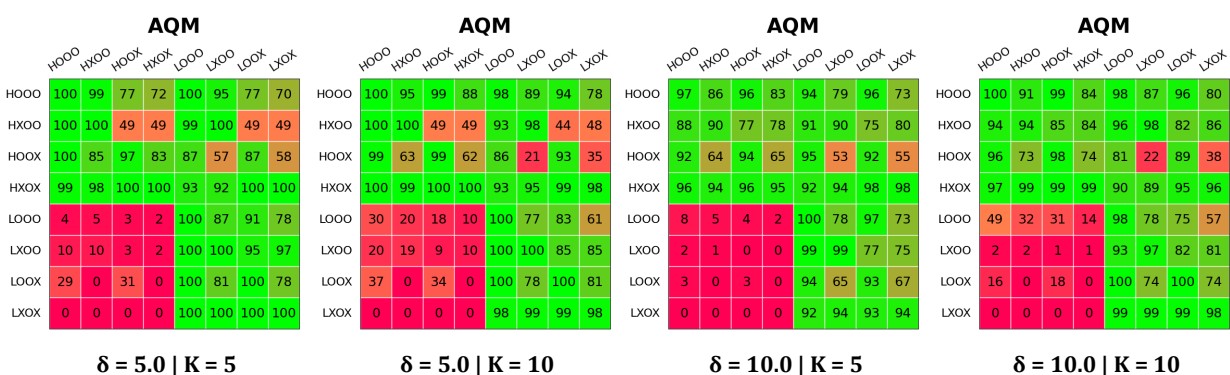

*Figure 14.* **AQM hyperparameter sensitivity.** Performance heatmaps of AQM as a function of the separation radius $\delta$ (controls keypoint sparsity) and the subgoal sampling distance $K$ (controls how far subgoals are sampled along trajectories). Rows denote the training environment and columns denote the evaluation environment.

# H. Learned State-Space Mappings

## H.1. OGBench – AntMaze

### H.1.1. ANALYSIS OF LEARNED MAPPINGS

*Figures 15*, *16*, and *17* visualize the topological graphs learned by **AQM** across the `medium`, `large`, and `giant` mazes. Specifically, they depict the spatial distribution of the selected keypoints (red dots) relative to the environment layout.

**Influence of the Repel Radius ($\delta$) :** As expected, we observe a strict inverse relationship between the repel radius and graph density. At $\delta = 10.0$ the algorithm produces a dense covering that captures granular connectivity and small paths. At $\delta = 25.0$ the graph is significantly pruned, retaining higher level topological representation of the maze. Crucially, regardless of the $\delta$ value, the spatial distribution remains uniform across the reachable manifold. The keypoints respect the environment's geometry solely based on the learned quasimetric and available transitions.

**Influence of the Dataset :** The mappings derived from `navigate` and `stitch` datasets exhibit highly similar densities and distributions. However, mappings generated from the `explore` datasets appear consistently denser, particularly at lower $\delta$ values. This can notably be attributed to the nature of the data : the `explore` dataset contains 5 times more transitions (5 M vs 1 M) and consists of random walks that cover the state-action space more exhaustively than the directed, task-oriented trajectories of the other datasets. This increased data density provides the dominating set algorithm with a larger pool of candidates, allowing it to fill minor topological gaps that might be sparsely represented in the expert datasets.

Interestingly enough, these visualizations underscore the potential of the generated graphs as diagnostic tools for exploration in Online RL or data collection in Offline RL. Because the keypoint distribution mirrors the support of the collected data, the topological structure effectively delineates the boundaries of the agent's knowledge. Thus, gaps in the graph may serve as explicit indicators of missing transitions or barriers. Following contemporary research (Zhang et al., 2025), this suggests a promising avenue for leveraging this *topological uncertainty* to guide active exploration, where an agent prioritizes data collection in regions where the manifold approximation is fragmented, thereby progressively refining the topological graph.

### H.1.2. OGBENCH – ANTMAZE – MEDIUM

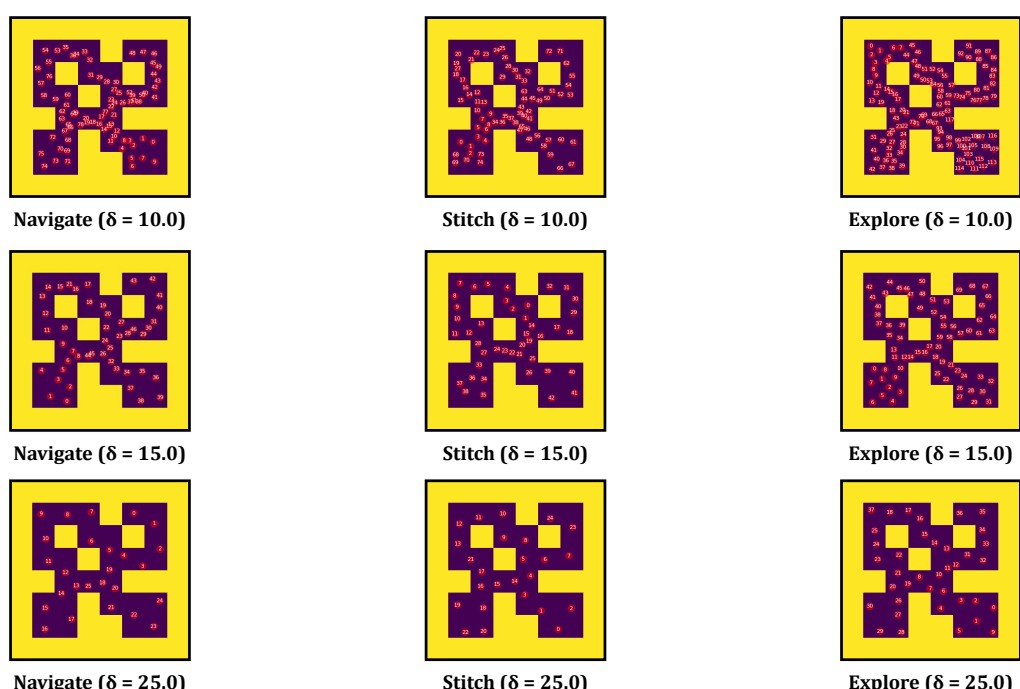

| Navigate ($\delta$ = 10.0) | Stitch ($\delta$ = 10.0) | Explore ($\delta$ = 10.0) |
| Navigate ($\delta$ = 15.0) | Stitch ($\delta$ = 15.0) | Explore ($\delta$ = 15.0) |
| Navigate ($\delta$ = 25.0) | Stitch ($\delta$ = 25.0) | Explore ($\delta$ = 25.0) |

*Figure 15.* **OGBench** : Learned Mappings on **Medium Maze** according to the Dataset and the Repel Range ($\delta$).

H.1.3. OGBench – AntMaze – Large

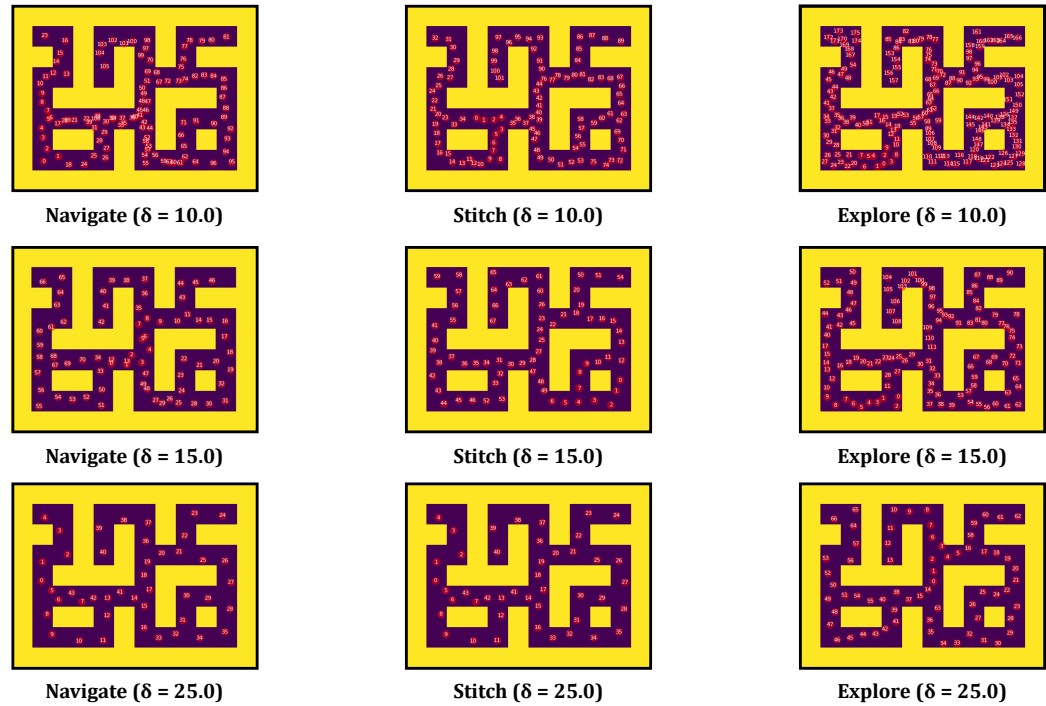

*Figure 16.* **OGBench** : Learned Mappings on **Large Maze** according to the Dataset and the Repel Range ($\delta$).

H.1.4. OGBench – AntMaze – Giant

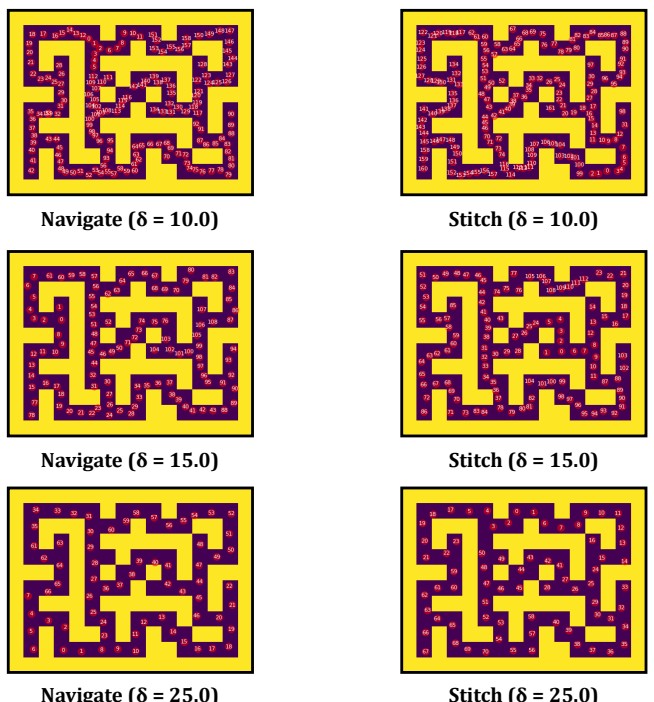

*Figure 17.* **OGBench** : Learned Mappings on **Giant Maze** according to the Dataset and the Repel Range ($\delta$).

## H.2. Continual NavBench – AmazeVille

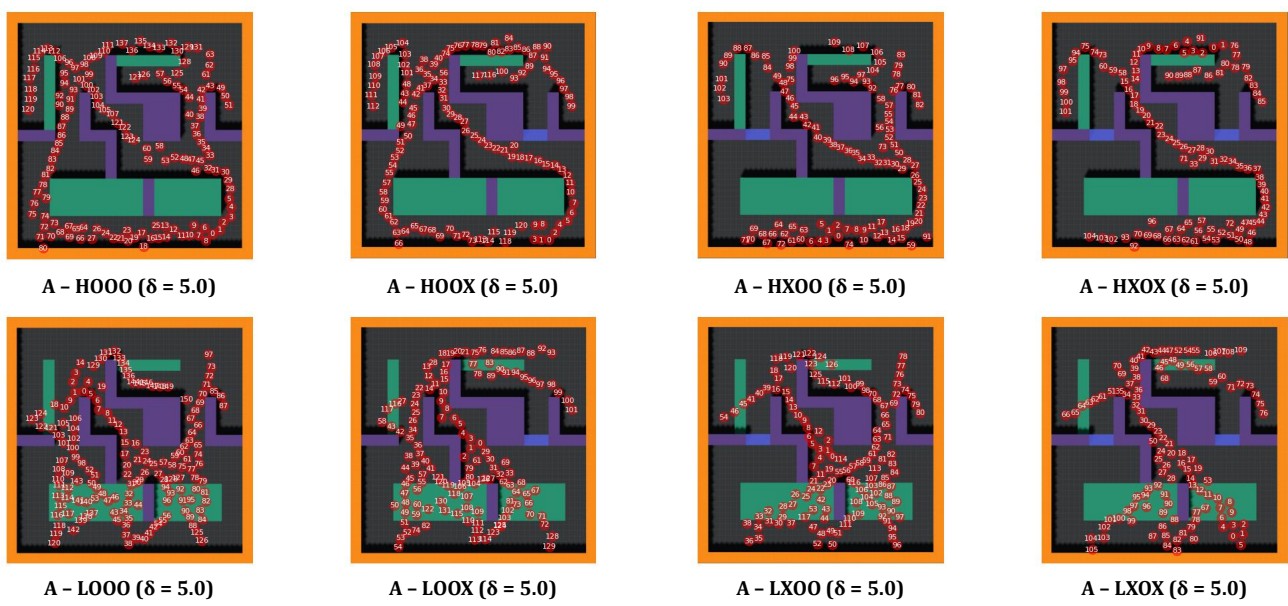

Figure 18. **Continual NavBench** : Learned Mappings on **AmazeVille** according to the Dataset and the Repel Range ($\delta = 5.0$).

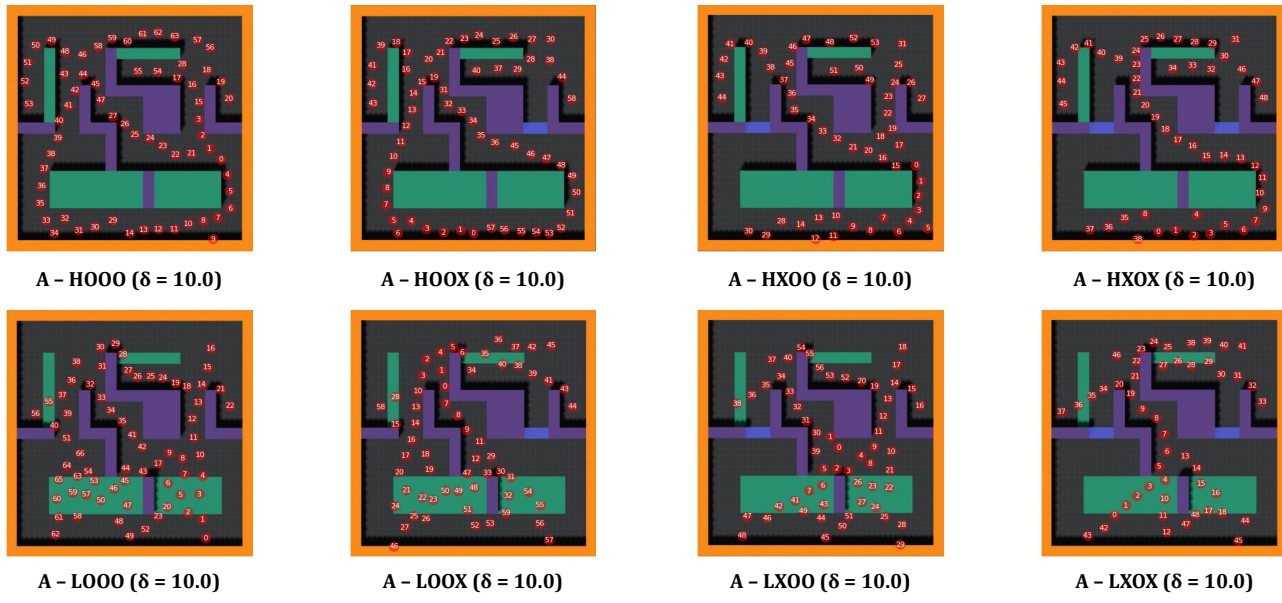

Figure 19. **Continual NavBench** : Learned Mappings on **AmazeVille** according to the Dataset and the Repel Range ($\delta = 10.0$).

