# OpenReview forum: "Adaptive Quasimetric Mapping : Principled Topological Abstraction for Robust Offline Goal-Conditioned Navigation"
_ICML.cc/2026/Conference — ICML 2026 regular_

### Official Review · Reviewer_xrgo · 2026-02-28

**Soundness:** 3
**Presentation:** 3
**Significance:** 2
**Originality:** 1
**Overall Recommendation:** 2
**Confidence:** 5

**Summary:**

This paper proposes Adaptive Quasimetric Mapping (AQM) for offline goal-conditioned long-horizon navigation. AQM first learns a time-to-reach quasimetric from offline data and then constructs a topological graph using a δ-separated keypoint selection strategy. Policy learning is performed using a quasimetric-based Subgoal Advantage-Weighted (SAW) method. At test time, the method monitors failures and removes unreliable edges from the graph to enable adaptive replanning. Experiments on OGBench AntMaze and Continual NavBench demonstrate state-of-the-art performance while requiring significantly fewer keypoints than prior graph-based methods.

**Compliance With Llm Reviewing Policy:**

Affirmed.

**Final Justification:**

Thank you to the authors for the thoughtful and detailed effort in carefully analyzing and comparing AQM with GAS in the rebuttal.

The authors attribute the improved sparsity primarily to the graph construction. However, an alternative explanation is that the stronger low-level policy (Q-SAW vs. TD3+BC) enables more reliable long-horizon stitching, thereby reducing the need for dense keypoints.

To support their claim, I believe the following experiments are necessary:

1) AQM graph + TD3+BC policy: to verify whether the proposed graph alone can maintain performance without SAW.
2) GAS graph + SAW policy: to test whether replacing the policy alone yields similar sparsity-performance trade-offs.
Without these ablations, it remains unclear whether the scaling advantage comes from the proposed graph abstraction or from the choice of policy.

For the budget-based test-time adaptation mechanism,
I think it seems to be more like a heuristic refinement than a standalone conceptual novelty. In particular, its spirit seems related to prior adaptive graph/planning approaches such as [BEAG], which also detects unattainable subgoals, removes unreliable edges, and updates the planning structure accordingly. While I agree that incorporating such a mechanism into an offline quasimetric graph framework is useful, I am less convinced that this component by itself constitutes a major research contribution.

[BEAG], Breadth-First Exploration on Adaptive Grid for Reinforcement Learning, ICML 2024.

**Key Questions For Authors:**

## Q1. Why Does AQM Require Fewer Graph Nodes than GAS?

The paper highlights that AQM achieves comparable or superior performance to GAS while using significantly fewer keypoints. However, it is unclear which component primarily enables this scaling advantage.

(1) Whether the improvement stems mainly from
* (a) Representation: Quasimetric vs. HILP in GAS
* (b) Graph Construction: Iterative Wavefront Expansion vs. Temporal Distance-aware Clustering in GAS
* (c) Policy Learning: Q-SAW  vs. TD3+BC in GAS

(2) Authors argue that deploying GAS requires up to 15,000 keypoints (e.g., on antmaze-giant-explore), framing this as a prohibitive computational burden. However, the GAS paper (Table 5) reports configurations with substantially fewer nodes (e.g., ~2,500 nodes) that already achieve strong performance, while the 15,000-node setting appears as a higher-density variant with only marginal performance gains. The phrasing “requires 15,000 keypoints” may overstate the practical requirements of GAS. I encourage the authors to revise this discussion to ensure a fair and accurate characterization of prior work.

## Q2. Question on the novelty of the keypoint selection algorithm

* The paper motivates the keypoint selection procedure through dominating set theory and maximal δ-separated sets. However, Algorithm 2 (Iterative Wavefront Expansion) appears structurally equivalent to a greedy δ-cover (or maximal separated set) construction in a learned latent distance space.

* This seems closely related to the uniform covering strategy employed in prior graph-based RL methods such as GAS, where nodes are selected to evenly cover a learned temporal distance representation space.

* In what precise sense does IWE differ algorithmically from prior greedy δ-cover constructions used in graph-based stitching methods?

## Q3. Applicability to Visual Observation Settings

The experiments appear to be conducted in low-dimensional state-based navigation environments (e.g., AntMaze, Continual NavBench). Could the authors clarify whether AQM has been evaluated, or is expected to generalize, to visual (pixel-based) observation settings?

(1) Is the quasimetric learned directly over full state vectors (e.g., including positions), or over selected features (e.g., x–y coordinates)?

(2) How would the δ-separated keypoint construction and time-to-reach calibration behave under high-dimensional visual embeddings?


## Q4. Scope of Evaluation Beyond Navigation

* Most of the compared baselines (e.g., flat, hierarchical, and graph-based offline RL methods) are designed to be generally applicable beyond navigation, including manipulation and other goal-conditioned control tasks. Could the authors clarify why the empirical evaluation focuses exclusively on navigation environments?

* Is AQM inherently tailored to navigation-style state spaces, or do the authors expect it to generalize to broader goal-conditioned tasks such as manipulation?

**Limitations:**

The authors discuss several technical limitations, particularly the dependence on dataset coverage and the restriction to deterministic environments. However, the discussion could be strengthened by more explicitly addressing the generalization limits to high-dimensional (e.g., visual) settings, the robustness of budget-based pruning under noisy dynamics, and potential safety implications of incorrect edge invalidation in real-world deployments. A broader reflection on societal impact, even if minimal, would improve completeness.

**Strengths And Weaknesses:**

## 1. Soundness

(1) Strengths:
* The paper appropriately cites existing theoretical results on quasimetric learning, including universal approximation and value function approximation properties. The δ-separated keypoint selection is well-formalized from a dominating set perspective. The test-time adaptation mechanism is clearly defined and logically consistent within the proposed framework.

(2) Weaknesses:
* The paper introduces limited new theoretical results beyond existing quasimetric RL foundations. While the Continual NavBench experiments are well-motivated, the adaptation scenarios remain relatively controlled, and the generalization to real-world levels of distribution shift appears limited.

## 2. Presentation

(1) Strengths:
* The overall structure of the paper is clear, and the methodological flow is logically organized.

(2) Weaknesses:
* The paper does not sufficiently highlight its core novelty. The unique contributions relative to prior graph-based navigation approaches are not articulated as sharply as they could be.

## 3. Significance

(1) Strengths:
* The scaling analysis—particularly the reduction in graph node counts compared to GAS—is practically meaningful and well-demonstrated.

(2) Weaknesses:
* The scope of the contribution is largely specialized to navigation, limiting its broader impact on general RL or the wider ML community. In Continual NavBench, the adaptation scenarios involve largely the same underlying graph structure with only gate open/close variations, which may not fully constitute substantial or realistic adaptation.

## 4. Originality

(1) Strengths:
* The integration of quasimetric learning into graph construction, along with the application of dominating set principles for keypoint selection, is conceptually well-motivated.

(2) Weaknesses:
* The contribution appears closer to a structured integration of existing ideas rather than a fundamentally novel methodological breakthrough.

---

> ### Author Rebuttal · Authors · 2026-03-31
>
> ## Introduction
> We thank the reviewer **R-xrgo** for the careful assessment and their feedback. Below we address each concern and indicate the clarifications we will make in the revision. We acknowledge our answers might be short due to the length limit, feel free to ask further details.
> ## W1 & W3 - Limited new theory beyond QRL, controlled adaptation benchmark, and specialized significance
> We respectfully disagree with the implication that these points make the contribution weak. The paper does not claim a new standalone quasimetric-learning theory beyond IQE/QRL [1, 2]. Its theoretical contribution is at the abstraction layer built on top of that foundation for navigation from offline data.
>
> We also agree that Continual NavBench is structured rather than open-ended. This is meant to isolate whether an offline graph-based agent can **detect topological invalidation and reroute within support** after the environment changes. In that sense, the benchmark is controlled for diagnosis, not because the problem is trivial. **The current results already show both the intended success mode and the key failure mode**: AQM can reroute under blocked-path changes when alternative feasible behaviors exist in the data, and can fail when the required detour behaviors are absent from the dataset. We will therefore state the claim more explicitly as **support-bounded topology adaptation**, not any-shift robustness (which is an open problem).
> ## W2 / W4 - Core novelty is not articulated sharply enough; originality appears to be only a structured integration of existing ideas
> We respectfully disagree with the assessment of originality. The present work’s contribution does not lie in proposing a wholly new method, but rather in the systematic and task-specific integration of existing approaches. The paper contributes:
>
> (1) a **new perspective** in which learned quasimetric geometry is used not only as a scoring signal, but as the basis for a sparse topological abstraction for offline navigation;
>
> (2) a **well-justified combination** of learned reachability geometry, sparse-cover graph construction, and budget-based edge invalidation / rerouting;
>
> (3) explicit cover and connectivity **guarantees** for that abstraction layer;
>
> (4) an **adaptive operating point** on long-horizon navigation.
>
> In graph-based offline GCRL, a scientifically relevant question is not only how to learn a signal, but **how to turn that signal into a usable abstraction/planning interface**, ideally with clear guarantees and deployment-relevant behavior. In consequence, we will update the contribution bullets.
> ## Q1 / Q2 - Why fewer nodes than GAS, and in what sense is IWE novel?
> On the GAS comparison, we agree that the wording “requires 15,000 keypoints” is too strong. We will revise this part to instead highlight GAS's 2500 keypoints for 94% perf. variant on the Giant maze, which is a better tradeoff. We note that this does not change our overall analysis, as **AQM is still significantly sparser in its graph representation**, e.g. 114 keypoints for 94% perf. on the Giant maze.
>
> For IWE specifically, we agree that at the algorithmic level it is structurally related to greedy maximal separated-set / δ-cover classical theory. **The novelty lies in the application, in the context of navigation from offline data**, while preserving theoretical guarantees [App. E].
> ## Q3 / Q4 - Applicability to visual settings and scope beyond navigation
> The experiments are not restricted to (x,y)-coordinates, we use full state inputs. In antmaze it is for both state and goal space. In Continual NavBench, the observation space includes current position, orientation, velocity, and a low-res depth image, with the goal space using (x,y,z).
>
> For visual settings multiple problems are stacked together (representation learning, hyperparameters, partial observability, dataset covering). Although the theoretical framework may hold, we do not claim that the current theory or experiments establish AQM in visual settings. **We argue that state-based adaptive navigation from offline data is a relevant setting by itself**. Nevertheless, pixel-based variants are an interesting future direction to consider
>
> Conceptually, the framework may extend beyond navigation when there exists a temporal geometry and a useful sparse abstraction. However, in manipulation domains the inputs expand with object pose configurations, which make the density assumptions harder to verify in practice. So we prefer not to overclaim. We will revise Section 6 to make this more explicit.
>
> ## Conclusion
> We hope these clarifications make the paper’s scope and contribution more precise. We would be glad to clarify any remaining points during the discussion phase.
>
> [1] Improved Representation of Asymmetrical Distances with Interval Quasimetric Embeddings. T. Wang et al. 2022
>
> [2] Optimal Goal-Reaching Reinforcement Learning via Quasimetric Learning. T. Wang et al. 2023

---

> > ### Author Rebuttal · Reviewer_xrgo · 2026-04-02
> >
> > Thank you for the clarification. However, the response still does not clearly identify the primary reason why AQM can maintain performance with substantially fewer keypoints than GAS. Since sparse graph scalability is one of the central claims of the paper, I believe this point requires a more precise analysis.
> >
> > In particular, I would strongly encourage the authors to provide a clearer causal decomposition, explaining whether the advantage mainly comes from
> > (a) the learned representation (quasimetric vs. HILP),
> > (b) the graph construction method (IWE vs. GAS graph construction), or
> > (c) the low-level policy learning method (Q-SAW vs. TD3+BC),

---

> > > ### Author Response · Authors · 2026-04-07
> > >
> > > We thank **R-xrgo** for their follow-up and thorough review, notably for pressing on the GAS comparison. It was genuinely useful and helped us to better understand GAS.
> > >
> > > The table below separates the main points of overlap and difference between AQM and GAS:
> > > | Aspect | AQM | GAS | Comparison | Relevancy |
> > > |---|---|---|---|---|
> > > | 1. Backbone | Quasimetric + Graph Planning + Low Policy | Temporal-Distance + Graph Planning + Low Policy | Both leverage graphs for efficient planning | GAS is the right baseline to compare AQM to |
> > > | 2. Distance | Interval Quasimetric Embeddings (IQE) approximating the minimum directed distance (QRL objective) | L2 Temporal Distance Representation (TDR) approximating the minimum distance (IQL objective) | Both leverage temporal distance to build graph, plan and extract policies | Going from A to B may not have the same cost as going from B to A. This emphasizes theoretical study, although neither AQM or GAS insists on that empirically |
> > > | 3. Filtering | No filtering of the data | Temporal-Efficiency filtering of the data at trajectory level | Only GAS filters out its trajectories before graph construction | Removing filtering in GAS reduces performances (see their own ablations). AQM does not need filtering |
> > > | 4. Graph Construction | IWE $\delta$-separated cover over the datase | TD-aware clustering in TDR space given a minimal threshold $H_{TD}$ | Both rely on a greedy keypoint selection process using a minimal distance threshold, and GAS does latent clustering | We will highlight this in the updated manuscript |
> > > | 5. Graph Support | Nodes of the graph are states from the dataset | Nodes of the graph are cluster centroids in the latent space | AQM is grounded in the data support at a node-level, GAS uses cluster centroids | This difference may explain the need for more keypoints in GAS (smaller $H_{TD}$ ensure less OOD but more keypoints) |
> > > | 6. Theoretical Grounding | Covering and connectivity study in quasimetric space, discussion on approximation errors | Graph construction is well motivated, with theoretical insights but without any analysis | Although both are system papers with layered mechanisms, AQM is theory guided with studies from data to mapping and connectivity | AQM’s graph compactness and connectivity are not only empirical, we state when the guarantees hold and how approximation error weakens them |
> > > | 7.  Scaling: | Compact graph with good performance (#nodes / success) : 158 / 88 on giant-navigate ; 163 / 90 on giant-stitch ; 114 / 94 on large-explore | Best reported results on the same tasks (#nodes / success) : 978 / 77 ; 1966 / 88 ; 2499 / 94 | Both are strong graph-based methods; AQM reaches competitive or better success with much smaller graphs | AQM offers a different success / adaptation operating point, with a practical graph efficiency |
> > > | 8. Low-level Controller | Q-SAW with independent subgoal sampling | DDPG+BC with subgoal sampling aligned with $H_{TD}$ | Both papers show that low-level control must align with the graph abstraction. In ablations AQM varies the controller, while GAS varies the subgoal-sampling rule | Graph construction largely explains compactness, while low-level policy explains performance (although linked in GAS) |
> > > | 9. Empirical Scope | Navigation-focused setting: Performance + Scalability + Adaptation | Performance only on broader scope : Locomotion + Navigation + Manipulation | GAS is broader, AQM is deeper on navigation with support-bounded adaptation | This supports a specialized significance of AQM in the regime it targets |
> > >
> > > From the evidence currently in the two papers, the primary driver of AQM’s smaller number of keypoints is the graph-construction, not the low-level policy. It separates the sparsity radius, from graph connectivity, and from low-level horizon choice. GAS instead builds a TD-aware clustering graph and an extracted policy around a shared temporal threshold $H_{TD}$​, and its own ablations show that making $H_{TD}$ larger reduces node count but often degrades performance.
> > >
> > > We respectfully disagree with the implication that this previously unresolved comparison exhausts the contribution. As we mentioned, AQM also contributes two elements that are separate from the node-count question:\
> > > **(1) a sparse-cover and connectivity theoretical analysis for the learned mappings** ;\
> > > **(2) a budget-based test-time adaptation mechanism for support-bounded topology change**.
> > >
> > > From the remaining question, namely what makes quasimetrics better than value functions beyond theory, we believe there is ground for future work. GCRL benchmarks rarely assess a distance in a *pure metric* way. They usually evaluate it through the extracted policy performance and abstraction it enables, meaning that a strong reachability signal can still look weak under the wrong abstraction.
> > >
> > > We hope this clearer positioning helps show why the paper makes a meaningful contribution, and we would be happy to provide any further detail if useful.

---

### Official Review · Reviewer_VyEV · 2026-03-12

**Soundness:** 3
**Presentation:** 3
**Significance:** 3
**Originality:** 3
**Overall Recommendation:** 4
**Confidence:** 3

**Summary:**

This paper proposes Adaptive Quasimetric Mapping (AQM) for offline goal-conditioned navigation. The core idea is to learn a time-to-reach
quasimetric function. The method uses this function to build a sparse graph of key keypoints. It performs planning using shortest-path search. It has three main parts.

First, It learns a distance function using Interval Quasimetric Embeddings (IQE).

Second, It selects keypoints using a greedy algorithm named Iterative Wavefront Expansion.

Last, during execution, it checks the real travel time of each edge. If the time and the predicted time-to-reach value do not match, the system will remove the edge and replanning.

**Compliance With Llm Reviewing Policy:**

Affirmed.

**Final Justification:**

Thank the authors for their detailed and well-structured rebuttal. The additional experiments and clarifications provided have adequately addressed my major concerns. I am satisfied with the responses and look forward to seeing the promised revisions incorporated into the final version of the paper.

**Key Questions For Authors:**

1 As I mentioned in the weakness, can the author conduct some experiments using the dataset that covers only part of the environment to validate the dataset coverage robustness of the method?

2 The authors should compared their method with Offline Goal-Conditioned RL with Quasimetric Representations (Myers et al., 2025). As the method is SOTA, is cited in your paper, and is open-sourced.

3 Some mistakes in equations should be corrected. For example: in Eq. (3), the multiplier $\lambda$ is included in the $L_{close}$. However, in Algorithm 1, line 8 of the for loop, $\lambda$ is used to multiply the $L_{close}$ again.

**Limitations:**

yes

**Strengths And Weaknesses:**

Strengths

1 This work uses time-to-reach quasimetric function for topological abstraction. It is a geometric prior for navigation tasks and helps build a connectivity graph which can be used for path planning

2 AQM constructs a sparse graph abstraction. It can help achieve better performance than GAS while significantly reducing the nodes and planning complexity.

3 The proposed method achieves SOTA performance against flat, hierarchical and graph-based Offline RL methods.

Weaknesses

1 The theory assumes that the learned quasimetric can approximate the real time-to-reach distance. However, this assumption may not hold in offline RL. Most of the time, the dataset covers only part of the environment.

2 Although the method claimed the best performance on the OGBench AntMaze suite, the performance improvement is not large compared to GAS. Most importantly, the paper does not compare with other SOTA baselines cited in this paper, such as TRL, TMD, and Offline Goal-Conditioned RL with Quasimetric Representations (Myers et al., 2025). And the last work is open-sourced.

---

> ### Author Rebuttal · Authors · 2026-03-31
>
> ## Introduction
> We thank reviewer **R-VyEV** for their thoughtful review and the proposed improvements. We appreciate that they highlighted the motivations of AQM and the results obtained. We address the questions and issues raised below and will revise the manuscript accordingly. Due to character limits our answers are brief. Please ask if you need more detail.
> ## W2 - Although the method claimed the best performance on the OGBench AntMaze suite, the performance improvement is not large compared to GAS.
> We partially agree, and we will state this more carefully in the paper.
>
> **If we only look at average success rate on OGBench, the gap over GAS [1] can indeed be modest**.
>
> **However, the intended claim of the paper is not that AQM uniformly dominates GAS on success rate alone**. The paper also targets an operating point. Indeed, we seek **long-horizon navigation with a computationally efficient graph, covering guarantees, and adaptability at test time**. This is notably how Section 5.2.2 motivates Table 2, as we argue that dense keypoint graphs create a practical burden because planning scales polynomially with graph size in terms of number of keypoints. It also explicitly frames this as a deployment issue in resource-constrained production environments, where efficiency often matters more than performance, and we propose both in our setting. Given that important context, the difference is empirically more meaningful, as AQM achieves comparable or better performance with significantly less keypoints.
> ## Q1 - Can the author conduct some experiments using the dataset that covers only part of the environment to validate the dataset coverage robustness of the method?
> We agree that this is an important concern. However, we do not think an additional synthetic experiment is necessary to establish the qualitative answer, because the paper already answers it in both the theory and experiments:
>
> **By construction, AQM is support-bounded**. Sec. 4.3 and App. E state the $(\delta+\varepsilon)$-cover result under explicit assumptions, including $\varepsilon$-density of the dataset support.
>
> **The paper already states the practical consequence of missing support**. App. E explicitly says that when data is sparse, holes appear in the support, planning quality may degrades, and AQM is expected to fail or require additional mechanisms such as exploration, data augmentation, or stronger priors.
>
> **The same phenomenon already appears empirically in Continual NavBench**. Section 5.2.3 shows that transfer from L-maps (maps with blocks we can jump on) to H-maps (maps with higher blocks we have to contour) fails when the training data do not contain the required detour behaviors. We state: “The graph cannot stitch a valid solution from the available transitions” and adaptation remains “strictly bounded by the behavioral coverage of the training data.”. This concerns both path planning and skills to perform.
>
> This is why we frame AQM as **adaptation within support**. This is not only a technical constraint but often a **safer and more usable design choice**. In fact, the goal is to reroute through already known areas, rather than extrapolate into unseen parts of the space.
> ## Q2 - Comparison to TMD [2]
>
> We agree that TMD (flat policy approach leveraging quasimetric, supporting stochastic environments) is a relevant baseline and we appreciate the suggestion. Our current comparison set covers main method families to the paper’s claims: flat ones, hierarchical ones, and graph-based ones. In that sense, the present evaluation already includes both a quasimetric-based flat baseline (QRL) and strong long-horizon flat baselines such as SAW, as well as the strongest graph-based baseline, GAS. In particular, SAW is the best-performing flat method on test environments shared between the TMD paper and ours.
>
> Moreover, since we consider stochastic environments as future work, it would become interesting to consider TMD as a baseline or even leverage the method to build topological graphs that encode stochasticity.
> ## Q3 - Some mistakes in equations should be corrected. For example: in Eq. (3), the multiplier $\lambda$ is included in the $L_{close}$. However, in Algorithm 1, line 8 of the for loop, $\lambda$ is used to multiply the $L_{close}$ again.
> We thank **R-VyEV** for their sharp reading which helps to improve the quality of our proposition. We agree that the mentioned notation is inconsistent,  and we will correct Alg. 1 at line 8 and Alg. 1 at line 10 accordingly, by removing the $\lambda$ before $L_{close}$.
> ## Conclusion
> We hope these updates provide a clearer picture of our work, and we are available to address any remaining questions.
>
> [1] Graph-Assisted Stitching for Offline Hierarchical Reinforcement Learning. S. Baek, T. Park, J. Park, S. Oh. Y. Kim. (2025)
>
> [2] Offline Goal Conditioned Reinforcement Learning with Temporal Distance Representations. V. Myers, B. C. Zheng, B. Eysenbach, S. Levine

---

> > ### Author Rebuttal · Reviewer_VyEV · 2026-04-02
> >
> > Thank the authors for their detailed and well-structured rebuttal. The additional experiments and clarifications provided have adequately addressed my major concerns. I am satisfied with the responses and look forward to seeing the promised revisions incorporated into the final version of the paper.

---

> > > ### Author Response · Authors · 2026-04-06
> > >
> > > We are very thankful to **R-VyEV** for their relevant remarks and the useful discussion and updates that followed. We are very glad that the additional clarifications and revisions addressed your main concerns and made the intended contribution of the paper clearer. We will make sure these points are reflected clearly in the final version, and we would of course be happy to clarify or improve anything further if useful during the final evaluation.

---

### Official Review · Reviewer_JCvf · 2026-03-13

**Soundness:** 4
**Presentation:** 4
**Significance:** 3
**Originality:** 4
**Overall Recommendation:** 5
**Confidence:** 2

**Summary:**

This paper presents a graph-inspired approach for Offline RL for navigation. The approach learns a quasimetric function that maps the distance or time-to-reach between states, builds a graph between these states, and finally trains a policy to reach a goal. The approach is compared to baselines on navigation benchmarks.

**Compliance With Llm Reviewing Policy:**

Affirmed.

**Final Justification:**

The discussion solidified my judgment that this paper is worthy of acceptance.

**Key Questions For Authors:**

1) Are there any rules of thumb to select the repel radius, or is this done for every environment?

2) How many samples are required to train this approach? How long does this take on the hardware used? How does that compare to the baselines?

3) (out of curiosity) Does this approach generalize to non-navigation tasks? An example would be long-horizon planning tasks, like robotic manipulation.

**Limitations:**

Yes.

**Strengths And Weaknesses:**

# Strengths
- The approach is well motivated and clearly presented
- The evaluations are clearly presented and the experiments are well structured. Appropriate baselines seem to be used.
- The ability to generate significantly more sparse graphs is very valuable
- Robustness/generalization is also very relevant to this task
- Selection of the hyperparameter for the repel radius is backed up with experimental data

# Weaknesses
- How dependent is the method on the accuracy of each component? If the quasimetric is not well-trained, how bad does the method perform?
- Some of these components are known individually. The combination seems novel to me, but I am not an expert in these areas.
- Code to reproduce results was not made available.

---

> ### Author Rebuttal · Authors · 2026-03-30
>
> ## Introduction
> We are thankful to reviewer **R-JCvf** for the constructive review and the positive appreciation of our work. We answer the raised issues below and will revise the manuscript accordingly. Our answers are brief due to length limits, feel free to ask us for more details.
> ## Q1 - Are there any rules of thumb to select the repel radius, or is this done for every environment?
> **The repel radius $\delta$ is a hyperparameter selected per environment through a sweep [App. D.2] rather than by a universal closed-form rule**, although we partially discuss this in App. E.7.
>
> We increase $\delta$ to reduce the graph size, but stop before it becomes too thin to support reliable longest-horizon navigation. Fig. 6 shows that increasing $\delta$ reduces keypoint counts across all tasks, while success remains largely stable on the medium and large mazes, and it degrades on the giant maze once the graph becomes too sparse. For a reasonable $\delta$, performance is bounded more by the agent’s ability to reach the suggested subgoals [Sec. 5.3.3] than by graph density itself.
> ## Q2.1 - How many samples are required to train this approach? How long does this take on the hardware used?
> **We use standard offline datasets associated with each benchmark. For AntMaze, the navigate datasets contain $10^6$ transitions and the explore datasets contain $5\cdot10^6$ transitions. For Continual NavBench, the datasets are human-generated and contain $100$ episodes of around $200$ transitions each per maze configuration.**
>
> On the optimization side, App. D.2 states that all policy networks are trained with batch size 1024, all experiments run for $10^6$ gradient steps, and training is performed on NVIDIA V100 GPUs. **Typical full training wall-clock time is about 2 hours on OGBench and about 4 hours on Continual NavBench**. For the video-game study, this is largely due to evaluation overhead with the game engine, rather than gradient computation.
> ## Q2.2 - How does that compare to the baselines?
> As reported in App. D.4, **for wall-clock runtime, we intentionally do not claim a direct cross-paper comparison**. Indeed, implementations span different frameworks, engineering choices, and heterogeneous hardware/software stacks, so simple timing would mix algorithmic effects with software artifacts. A fair runtime study would require a unified setup, which is out of scope for the current submission [1, 2].
>
> **What the paper does support directly is the algorithmic / graph-size efficiency of the resulting abstraction**. Path planning complexity for graph-based methods scales polynomially in the number of keypoints. This is why the sparse-graph property is practically important as it affects not only one-shot path computation, but also repeated planning, localization, and online graph maintenance during adaptation.
> ## Q3 - (out of curiosity) Does this approach generalize to non-navigation tasks? An example would be long-horizon planning tasks, like robotic manipulation.
> The careful answer is **this is conceptually plausible, but not demonstrated in the current paper**.
>
> **The present empirical scope is navigation**, because this is a regime where time-to-reach geometry is meaningful, topological abstraction is easy to interpret, and topology-change adaptation can be evaluated cleanly. This is also a regime where the current theory aligns with the support/density assumptions used in Sec. 4.3 and App. E, which are easier to state and reason about in navigation than in object-interaction settings.
>
> **Conceptually, the framework could extend beyond navigation** when three conditions hold:
> (1) there is a meaningful time-to-reach geometry over states or goals ;
> (2) the task admits a useful sparse topological abstraction with informative intermediate states or modes ;
> (3) a local goal-conditioned controller can reliably execute transitions between nearby key states.
>
> **However, we do not want to overclaim here**. In richer manipulation settings, the relevant configuration space expands with object poses and contact configurations, so the support and density assumptions become much harder to justify and verify in practice.
> ## Conclusion
> We hope these clarifications make our proposition more precise and will enable a more confident assessment of our work.
> We would be happy to clarify any remaining concerns or questions during the discussion phase.
>
> [1] Performance comparison of deep learning frameworks. M. Yapici et al. 2021
>
> [2] Algorithm Runtime Prediction: Methods & Evaluation. F. Hutter et al. 2013

---

> > ### Author Rebuttal · Reviewer_JCvf · 2026-04-02
> >
> > I thank the authors for the rebuttal. My concerns have been addressed. While I appreciate the authors' desire for a change in review confidence, I do not feel comfortable over-representing my competence in this area of research. It seems clear to me that the arguments in the paper are well-structured, and evidence is provided to support the key claims.

---

> > > ### Author Response · Authors · 2026-04-06
> > >
> > > We are very thankful to **R-JCvf** for the update and for the careful discussion. We truly appreciate the time and attention you gave to the paper, and we fully understand your position. We are glad that the rebuttal helped make the structure of the work and the support for its main claims clearer. If any further questions arise at any stage, we would be very happy to clarify them.

---

### Official Review · Reviewer_QxMv · 2026-03-13

**Soundness:** 3
**Presentation:** 3
**Significance:** 2
**Originality:** 3
**Overall Recommendation:** 4
**Confidence:** 2

**Summary:**

This paper proposes an offline planning framework based on a learned time-to-reach quasimetric. The method constructs a sparse set of keypoints from the dataset using a greedy approximation to a dominating set, forming a topological abstraction of the environment. During inference, the agent performs planning over this graph and prunes edges using a traversal time budget derived from the learned quasimetric, enabling zero-shot replanning without additional training.

**Compliance With Llm Reviewing Policy:**

Affirmed.

**Final Justification:**

Thanks to the authors for the detailed rebuttal and the clarifications, which helped better position the paper and clarify its contribution.

**Key Questions For Authors:**

1. How is the time-to-reach quasimetric trained in practice? In particular, how are the ground-truth time-to-reach values obtained from the dataset? Is the obtained ground truth shortest path?
2. How sensitive is the performance to the number of selected keypoints in the sparse cover?
3. How does the computational cost of graph construction and planning scale as the environment size increases?

**Limitations:**

The proposed approach depends on the coverage of the offline dataset and may struggle in environments that are poorly represented in the training data. In addition, the method relies on constructing a sparse topological graph from the dataset, which may limit its applicability in highly dynamic or previously unseen environments.

**Strengths And Weaknesses:**

strength
1. The paper introduces an interesting formulation for goal-conditioned navigation using a learned time-to-reach quasimetric.
2. The paper provides detailed descriptions of the proposed method and experimental setup, which helps improve the clarity of the work.
3. The authors provide open-source code, which facilitates reproducibility and further research based on this work.

weakness
1. The necessity of the quasimetric formulation is not sufficiently validated. The paper does not provide a clear comparison between the quasimetric and standard metric-based distance functions.
2. The approach relies heavily on the coverage of the offline dataset. If the dataset does not sufficiently cover the environment, the constructed keypoint graph may fail to support effective planning.
3. The hyperparameter δ appears to play an important role in constructing the sparse cover, but the paper provides limited analysis on its sensitivity.

---

> ### Author Rebuttal · Authors · 2026-03-30
>
> ## Introduction
> We thank **R-QxMv** for the careful feedback and for finding our proposition clear. We address below the questions and remarks they raised and will revise the manuscript accordingly. Due to length limits, our answers are brief. Feel free to ask for more detail.
>
> An important clarification is that **AQM does not introduce a new quasimetric learner:
> it builds graph structures from a quasimetric**.
> Since studying topological objects often requires metric functions [1, 2], we leverage IQE and QRL [3, 4, Sec. 4.2] to learn such a function, with our main contributions being:
> (1) **sparse topological abstraction with covering guarantees** ; (2) **test time adaptation through budget-based change point detection**.
>
> ## Q1.1 - How is time-to-reach quasimetric learned in practice?
> In practice **we use IQE as the quasimetric function class and QRL for the calibration objective** with:
> - a local consistency loss $L_{close}$​ calibrating one-step transitions to have a distance of 1.
> - a global separation loss $L_{push}$​ preventing collapse by pushing states and goals apart.
>
> This formulation comes with optimality guarantees [Sec. 4.2]. See App. D.2. for implementation details.
>
> ## Q1.2-1.3 - How are the ground-truth time-to-reach values obtained from the dataset? Is the obtained ground truth shortest path?
> **We do not train the quasimetric with a supervised regression to shortest-path target values**. Instead we follow the aforementioned semi-supervised optimization objective (Sec. 4.2, App. D.3).
>
> **Regarding shortest-path, this notion appears instead at the planning stage**
> [Sec. 3, 4.3, 4.5] as we rely on Floyd-Warshall using a pre-computed distance matrix between keypoints.
>
> ## Q2 - How sensitive is the performance to the number of selected keypoints in the sparse cover?
> **The number of selected keypoints depends on the repel radius** $\delta$ [5, 6]. Sec. 5.3.2 shows a sparsity-performance trade-off controlled by $\delta$. As its value increases, the number of keypoints drops sharply. Over the studied range, success remains largely stable on medium and large mazes, while giant mazes benefit higher keypoint count.
>
> ## Q3 - How does the computational cost of graph construction and planning scale as the environment size increases?
> **Graph construction complexity and inference cost scale polynomially with the number of keypoints and, as environments become larger or longer-horizon, methods that require denser graph coverage incur rapidly increasing planning and localization costs.**
>
> We use Floyd-Warshall as a shortest-path solution. With $|Z|$ the number of keypoints, the pathfinding cost scales as $O(|Z|^3)$, as we compute a next-node look-up matrix at the initialization of an episode and when replanning during adaptation. Then for each step of an episode,  assigning the current state to the nearest keypoint scales linearly with $Z$. Empirically, Table 2 shows that AQM operates with much smaller graphs while remaining competitive.
>
> ## L1 - The proposed approach depends on the coverage of the offline dataset and may struggle in environments that are poorly represented in the training data.
> As in many offline learning algorithms, we acknowledge the data coverage issue is a common limitation. Since we believe this is a relevant point to mention, we made it explicit to carefully bound the scope of our research, we discussed the guarantees, and its practical limits are analyzed in App. E. Moreover, we proposed a relevant adaptation strategy to handle changes at test-time.
>
> ## L2 - The method relies on constructing a sparse topological graph from the dataset, which may limit its applicability in highly dynamic or previously unseen environments.
> While we partially agree, we believe the limitation is narrower than that wording suggests.
>
> AQM is designed for **adaptation within support**, not for arbitrary generalization. When we require behaviors beyond the dataset support, pruning/replanning alone is insufficient. General adaptation at test-time remains an open problem and proposed solutions, such as ours, require specifying a scope to make it feasible. We focused on a case that supports real life applications where dynamics or actions of the agent (autonomous vehicle, video game bot) don't change.
>
> ## Conclusion
> We hope these clarifications make our proposition more precise, and we would be glad to clarify remaining issues during the discussion phase.
>
> [1] An introduction to the geometry of metric spaces. S. Semmes. 2007
>
> [2] Learning The Minimum Action Distance. L. Steccanella et al. 2025
>
> [3] Improved Representation of Asymmetrical Distances with Interval Quasimetric Embeddings. T. Wang et al. 2022
>
> [4] Optimal Goal-Reaching Reinforcement Learning via Quasimetric Learning. T. Wang et al. 2023
>
> [5] Covering Compact Metric Spaces Greedily. J. Rolfes et al. 2018
>
> [6] Products of Metric Spaces, Covering Numbers, Packing Numbers and Characterizations of Ultrametric Spaces. O. Dovgoshe et al. 2009

---

> > ### Author Rebuttal · Reviewer_QxMv · 2026-04-03
> >
> > Thank you for the clarification. The rebuttal helps make the paper’s positioning clearer. In particular, the authors describe the contribution as lying in the systematic and task-specific integration of existing approaches, and further note that the method requires specifying a scope to make it feasible. This is helpful, but it also raises some questions for me regarding the novelty and generality of the proposed framework.
> >
> > In addition, I would appreciate more discussion of the practical applicability of the proposed framework. The rebuttal acknowledges that data coverage is a common limitation, but it remains unclear to me how this issue would be handled in more realistic deployment scenarios, such as autonomous vehicles, where environment changes and incomplete support are common.

---

> > > ### Author Response · Authors · 2026-04-06
> > >
> > > We thank **R-QxMv** for their follow-up. Raising a helpful question, as it pushes for a more precise explanation of AQM operating regime.
> > >
> > > Let us clarify the general-scope first. Describing AQM as a systematic and task-specific integration is accurate, but incomplete. The contribution is neither a new distance learner in isolation, nor a claim of general out-of-support adaptation, as we mention in our introduction of the setting. The contribution is the **theoretically grounded abstraction layer** we build, and the use we make of it **at test time** for **change detection** and **adaptation**.
> > >
> > > On practical applicability, AQM is a **support-bounded adaptation method from offline data**. It targets settings where the environment may change in the state space (for example, some routes become blocked or traversability changes), while the dynamics and action space remain unchanged, and the agent is expected to find an alternative route through already known feasible regions. If, after an environment change, there still exists a feasible route to the goal inside the covered support of the offline data, AQM can in principle detect failed edges and reroute. If all feasible post-change routes require states or behaviors absent from the offline support, then *no purely offline replanner* over that support *can guarantee recover* them without additional data. That is exactly the boundary encoded by the density assumption behind our graph guarantees, and it is also the failure mode we already observe in Continual NavBench when the required detour behaviors are missing from the dataset.
> > >
> > > This regime is practically relevant. Offline methods are attractive when exploration is costly or unsafe, the environment is largely known, and large logged datasets exist but are imperfect or suboptimal. In on-road autonomous cars, for example, one usually wants to stay on mapped roads rather than extrapolate into unsupported regions such as off-road areas. If an entire required route is absent from the data, trusting a purely offline planner there would be undesirable regardless of the specific method (going offroad). In telemetry-rich video game settings, maps may change locally due to player actions or world events, while large offline logs remain available. AQM is designed for such use cases.
> > >
> > > A more formal way to say the same thing is: our graph results assume a compact, path-connected support covered by the dataset up to $\varepsilon$ and approximated by the learned quasimetric. If an environment change effectively splits the reachable support into disconnected components and the start and goal lie in different ones, then the post-change task no longer satisfies those assumptions, and AQM is not expected to succeed. This is not a hidden weakness, we assume it and state it clearly, this is the intended boundary of the problem we target.
> > >
> > > The harder regime is when the dynamics or action space change, or when adaptation requires entering genuinely unseen regions. In that case, the offline assumption itself is no longer sufficient, and additional mechanisms such as map updates, online data collection, or exploration are needed.
> > >
> > > If adaptation is required anyway in a region with no data, then the offline assumption itself is no longer sufficient. The natural extension is an offline-to-online loop: use AQM as an initialization over the known support, perform safe or uncertainty-aware exploration near the boundary of that support, update the quasimetric with the newly collected transitions, and rebuild or repair the graph accordingly. That direction is consistent with offline-to-online RL methods such as balanced replay with pessimistic initialization [1], and with frontier-based exploration / map-update methods in robotics [2]. It is an important regime, but it is no longer *offline adaptation only*, it requires new data collection by design.
> > >
> > > We hope these clarifications make it easier to appreciate both the positioning and the contribution of the paper. If there are still questions or points that would benefit from further clarification, we would be glad to address them.
> > >
> > > [1] Offline-to-Online Reinforcement Learning via Balanced Replay and Pessimistic Q-Ensemble. Lee et al. 2021
> > >
> > > [2] A Graph-Based Reinforcement Learning Approach with Frontier Potential Based Reward for Safe Cluttered Environment Exploration. Calzolari et al. 2025

---

### Decision · Program_Chairs · 2026-04-30

**Decision:**

Accept (regular)

**Comment:**

This paper proposes offline graph-based navigation using learned time-to-reach quasimetrics and sparse topological abstraction with test-time adaptation. There are 3 positive reviews (1 strong accept + 2 weak accepts, confidence 2-3) vs. 1 negative review (strong reject, confidence 5). Strengths: practical 100x keypoint reduction, clear presentation, theoretical covering guarantees. Core weakness (reviewer: xrgo, high confidence): sparsity gains attributed to graph construction but causal ablations missing (lacks AQM+TD3+BC and GAS+SAW experiments).  The reviewers wonders whether the improvements could be due to policy instead. Additional concerns: quasimetric necessity not empirically validated, scope limited to navigation, test-time adaptation mechanism viewed as incremental heuristic. Rebuttal highly responsive but does not directly address main causal attribution concern. Overall, I suggest for this paper "WEAK ACCEPT", since three reviewers are positive and one is negative.